# Self-Improving Skill Learning for Robust Skill-based Meta-Reinforcement Learning

**Sanghyeon Lee, Sangjun Bae, Yisak Park, Seungyul Han**[*]
Graduate School of Artificial Intelligence
Ulsan National Institute of Science and Technology (UNIST)
Ulsan, South Korea 44919
{sanghyeon,bsjuntiger,isaac1018,syhan}@unist.ac.kr

## Abstract

Meta-reinforcement learning (Meta-RL) facilitates rapid adaptation to unseen tasks but faces challenges in long-horizon environments. Skill-based approaches tackle this by decomposing state-action sequences into reusable skills and employing hierarchical decision-making. However, these methods are highly susceptible to noisy offline demonstrations, leading to unstable skill learning and degraded performance. To address this, we propose Self-Improving Skill Learning (SISL), which performs self-guided skill refinement using decoupled high-level and skill improvement policies, while applying skill prioritization via maximum return relabeling to focus updates on task-relevant trajectories, resulting in robust and stable adaptation even under noisy and suboptimal data. By mitigating the effect of noise, SISL achieves reliable skill learning and consistently outperforms other skill-based meta-RL methods on diverse long-horizon tasks. Our code is available at https://github.com/epsilog/SISL.

## 1 Introduction

Reinforcement Learning (RL) has achieved significant success in various domains (Mnih et al., 2015; Andrychowicz et al., 2020). While advanced exploration frameworks have been proposed to improve sample efficiency (Pathak et al., 2017; Fortunato et al., 2018; Han & Sung, 2021a;b; Jo et al., 2024), traditional RL struggles to adapt quickly to new tasks. Meta-RL addresses this limitation by enabling rapid adaptation to unseen tasks through meta-learning how policies solve problems (Duan et al., 2016; Finn et al., 2017). Among various approaches, context-based meta-RL stands out for its ability to represent similar tasks with analogous contexts and leverage this information in the policy, facilitating quick adaptation to new tasks (Rakelly et al., 2019; Zintgraf et al., 2019; Kim et al., 2025). Notably, PEARL (Rakelly et al., 2019) has been widely studied for its high sample efficiency, achieved through off-policy learning, which allows for the reuse of previous samples. Despite these strengths, existing meta-RL methods face challenges in long-horizon environments, where extracting meaningful context information becomes difficult, hindering effective learning.

Skill-based approaches address these challenges by breaking down long state-action sequences into reusable skills, facilitating hierarchical decision-making and enhancing efficiency in complex tasks (Pertsch et al., 2021; 2022; Shi et al., 2023). Among these, SPiRL (Pertsch et al., 2021) defines skills as temporal abstractions of actions, employing them as low-level policies within a hierarchical framework to achieve success in long-horizon tasks. SiMPL (Nam et al., 2022) builds on this by extending skill learning to meta-RL, using offline expert data to train skills and a context-based high-level policy for task-specific skill selection. Despite these advancements, such methods are highly susceptible to noisy offline demonstrations, which can destabilize skill learning and reduce reliability. In real-world settings, noise often stems from factors like infrastructure aging or environmental perturbations, making it crucial to design methods that remain robust under such conditions (Brys et al., 2015; Chae et al., 2022; Yu et al., 2024a; Lee et al., 2025).

While noisy demonstration handling has been explored in other RL settings (Sasaki & Yamashina, 2020; Mandlekar et al., 2022), skill-based meta-RL has largely overlooked this challenge. We iden-

---

[*]indicates the corresponding author: Seungyul Han.

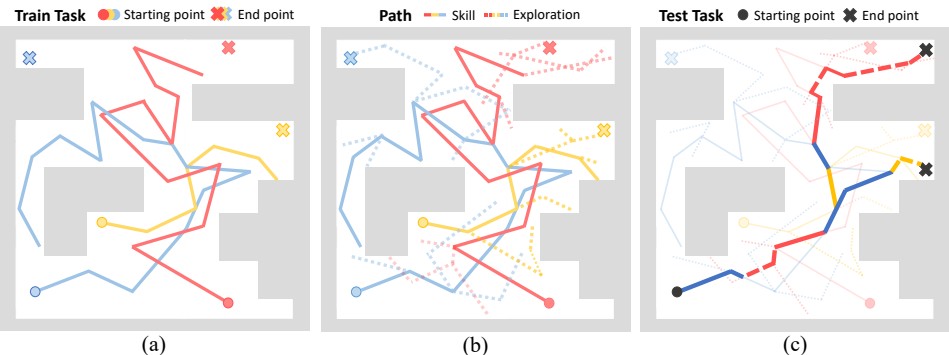

Figure 1: Sample trajectories in the Maze2D environment: (a) Noisy demonstrations from the offline dataset, (b) Trajectories explored by the exploration policy near the noisy dataset to uncover useful skills, and (c) Trajectories utilizing refined skills to solve unseen test tasks

tify a critical failure mode: when offline data are suboptimal, the skill library becomes corrupted, and this degradation propagates to the high-level policy, ultimately harming adaptation performance. To address this, we propose Self-Improving Skill Learning (SISL), a robust skill-based meta-RL framework with two key contributions: (1) Decoupled skill self-improvement, using an improvement policy that perturbs trajectories near the offline data distribution, discovers higher-quality rollouts, and self-supervises updates via a prioritized online buffer. This process progressively denoises the skill library while preserving stability. (2) Skill prioritization via maximum return relabeling, which evaluates offline trajectories with a learned reward model, assigns task-relevant hypothetical returns, and reweights them through a softmax prioritization scheme. This suppresses noisy or irrelevant samples and focuses skill updates on the most beneficial trajectories for adaptation. These components dynamically balance offline and online data contributions, yielding a progressively cleaner skill library and accelerating meta-RL convergence. To our knowledge, SISL is the first framework to explicitly address suboptimal demonstrations in skill-based meta-RL, significantly improving robustness and generalization of skill-learning in real-world noisy scenarios.

Fig. 1 illustrates how the proposed algorithm learns effective skills from noisy demonstrations in the Maze2D environment, where the agent starts at a designated point and must reach an endpoint for each task. Fig. 1(a) shows noisy offline trajectories, which fail to produce effective skills when used directly. In contrast, Fig. 1(b) demonstrates how the prioritized refinement framework uses the improvement policy to navigate near noisy trajectories, identifying paths critical for solving long-horizon tasks and refining useful skills through prioritization. Finally, Fig. 1(c) shows how the high-level policy applies these refined skills to successfully solve unseen tasks. These results highlight the method's ability to refine and prioritize skills from noisy datasets, ensuring stable learning and enabling the resolution of long-horizon tasks in unseen environments. This paper is organized as follows: Section 3 provides an overview of meta-RL and skill learning, Section 4 details the proposed framework, and Section 5 presents experimental results showcasing the framework's robustness and effectiveness, along with an ablation study of key components.

## 2 RELATED WORKS

**Skill-based Reinforcement Learning:** Skill-based RL has gained traction for tackling complex tasks by leveraging temporally extended actions. Researchers have proposed information-theoretic approaches to discover diverse and predictable skills (Gregor et al., 2016; Eysenbach et al., 2018; Achiam et al., 2018; Sharma et al., 2019), with recent work improving skill quality through additional constraints and objectives (Strouse et al., 2022; Park et al., 2022; 2023; Hu et al., 2024). In offline scenarios, approaches focus on learning transferable behavior priors and hierarchical skills from demonstration data (Pertsch et al., 2021; 2022; Shi et al., 2023; Xu et al., 2022; Kipf et al., 2019; Rana et al., 2023). Building upon these foundations, various skill-based meta-RL approaches have been developed, from hierarchical and embedding-based methods (Nam et al., 2022; Chien & Lai, 2023; Cho & Sun, 2024) to task decomposition strategies (Yoo et al., 2022; He et al., 2024) and unsupervised frameworks (Gupta et al., 2018; Jabri et al., 2019; Shin et al., 2024). For clarity, we use skill in the canonical skill-based RL sense: a latent over fixed-length action sequences. Therefore, studies (Yu et al., 2024b; Wu et al., 2019) that lack explicit skill learning are not categorized here.

**Relabeling Techniques for Meta-RL:** Recent developments in meta-RL have introduced various relabeling techniques to enhance sample efficiency and task generalization (Pong et al., 2022; Jiang et al., 2023). Goal relabeling approaches have extended hindsight experience replay to meta-learning contexts (Packer et al., 2021; Wan et al., 2021), enabling agents to learn from failed attempts. For reward relabeling, model-based approaches have been proposed to relabel experiences across different tasks (Mendonca et al., 2020), improving adaptation to out-of-distribution scenarios. Beyond these categories, some methods have introduced innovative relabeling strategies using contrastive learning (Yuan & Lu, 2022; Zhou et al., 2024) and metric-based approaches (Li et al., 2020) to create robust task representations in offline settings.

**Hierarchical Frameworks:** Hierarchical approaches in RL have been pivotal for solving long-horizon tasks, including goal-conditioned learning (Levy et al., 2019; Li et al., 2019; Gehring et al., 2021), and option-based frameworks (Bacon et al., 2017; Riemer et al., 2018; Barreto et al., 2019; Araki et al., 2021). The integration of hierarchical frameworks with meta-RL has shown potential for rapid task adaptation and complexity handling (Frans et al., 2018; Fu et al., 2020a; 2023). Recent work has demonstrated that hierarchical architectures in meta-RL can provide theoretical guarantees for learning optimal policies (Chua et al., 2023) and achieve efficient learning through transformer-based architectures (Shala et al., 2024). Recent advances in goal-conditioned RL have focused on improving sample efficiency (Robert et al., 2024; Hwang et al., 2025), state representation (Yin et al., 2024), and offline-to-online RL (Park et al., 2024; Schmidt et al., 2025). Offline-to-online RL assumes reward-annotated offline datasets to pretrain the policy based on RL before online fine-tuning. In contrast, our setting provides only reward-free offline data for skill learning, making direct application of offline-to-online RL infeasible and clearly distinguishing our approach.

## 3  BACKGROUND

**Meta-Reinforcement Learning Setup:** In meta-RL, each task $\mathcal{T}$ is sampled from a distribution $p(\mathcal{T})$ and defined as an MDP environment $\mathcal{M}^{\mathcal{T}} = \left(\mathcal{S}, \mathcal{A}, R^{\mathcal{T}}, P^{\mathcal{T}}, \gamma\right)$, where $\mathcal{S} \times \mathcal{A}$ represents the state-action space, $R^{\mathcal{T}}$ is the reward function, $P^{\mathcal{T}}$ denotes the state transition probability, and $\gamma$ is the discount factor. At each step $t$, the agent selects an action $a_t$ via the policy $\pi$, receives a reward $r_t := R^{\mathcal{T}}(s_t, a_t)$, and transitions to $s_{t+1} \sim P^{\mathcal{T}}(\cdot|s_t, a_t)$. The goal of meta-RL is to train $\pi$ to maximize the return $G = \sum_t \gamma^t r_t$ on the training task set $\mathcal{M}_{\text{train}}$ while enabling rapid adaptation to unseen test tasks in $\mathcal{M}_{\text{test}}$, where $\mathcal{M}_{\text{train}} \cap \mathcal{M}_{\text{test}} = \emptyset$.

**Offline Dataset and Skill Learning:** To address long-horizon tasks, skill learning from an offline dataset $\mathcal{B}_{\text{off}} := \{\tilde{\tau}_{0:H}\}$ is considered, which comprises sample trajectories $\tilde{\tau}_{t:t+k} := (s_t, a_t, \cdots, s_{t+k})$ without reward information, where $H$ is the episode length. The dataset $\mathcal{B}_{\text{off}}$ is typically collected through human interactions or pretrained policies. Among various skill learning methods, SPiRL (Pertsch et al., 2021) focuses on learning a reusable low-level policy $\pi_l$, using $q(\cdot|\tilde{\tau}_{t:t+H_s})$ as a skill encoder to extract the skill latent $z$ by minimizing the following loss function:

$$\mathbb{E}_{\substack{\tilde{\tau}_{t:t+H_s} \sim \mathcal{B}_{\text{off}}, \\ z \sim q(\cdot|\tilde{\tau}_{t:t+H_s})}} \left[\mathcal{L}(\pi_l, q, p, z)\right], \tag{1}$$

where $\mathcal{L}(\pi_l, q, p, z) := -\sum_{k=t}^{t+H_s-1} \log \pi_l(a_k|s_k, z) + \lambda_l^{\text{kld}} \mathcal{D}_{\text{KL}}(q||\mathcal{N}(\mathbf{0}, \mathbf{I})) + \mathcal{D}_{\text{KL}}(\lfloor q \rfloor||p)$, $H_s$ is the skill length, $\lambda_l^{\text{kld}}$ is the coefficient for KL divergence (KLD) $\mathcal{D}_{\text{KL}}$, $\lfloor \cdot \rfloor$ is the stop gradient operator, and $\mathcal{N}(\boldsymbol{\mu}, \boldsymbol{\Sigma})$ represents a Normal distribution with mean $\boldsymbol{\mu}$ and covariance matrix $\boldsymbol{\Sigma}$. Here, $p(z|s_t)$ is the skill prior to obtain the skill distribution $z$ for a given state $s_t$ directly. Using the learned skill policy $\pi_l$, the high-level policy $\pi_h$ is trained within a hierarchical framework using RL methods. In our paper, the skill refinement procedure builds on Eq. (1) and is applied during both initial skill learning phase and subsequent refinement. A more detailed description is provided in Appendix B.1.

**Skill-based Meta-Reinforcement Learning:** SiMPL (Nam et al., 2022) integrates skill learning into meta-RL by utilizing an offline dataset of expert demonstrations across various tasks. The skill policy $\pi_l$ is trained via SPiRL, while a task encoder $q_e$ extracts the task latent $e^{\mathcal{T}} \sim q_e$ using the PEARL (Rakelly et al., 2019) framework, a widely-used meta-RL method. During meta-training, the high-level policy $\pi_h(z|s, e^{\mathcal{T}})$ selects a skill latent $z$ and executes the skill policy $\pi_l(a|s, z)$ for $H_s$ time steps, optimizing $\pi_h$ to maximize the return for each task $\mathcal{T}$ as:

$$\min_{\pi_h} \mathbb{E}_{\tau_h^{\mathcal{T}} \sim \mathcal{B}_h^{\mathcal{T}}, e^{\mathcal{T}} \sim q_e(\cdot|c^{\mathcal{T}})} \left[\mathcal{L}_h^{\text{RL}}(\pi_h) + \lambda_h^{\text{kld}} \mathcal{D}_{\text{KL}}(\pi_h \| p)\right], \tag{2}$$

where $\lambda_h^{\text{kld}}$ is the KL divergence coefficient, $c^{\mathcal{T}}$ represents the contexts of high-level trajectories $\tau_h^{\mathcal{T}} := (s_0, z_0, \sum_{t=0}^{H_s-1} r_t, s_{H_s}, z_{H_s}, \sum_{t=H_s}^{2H_s-1} r_t, \cdots)$ for task $\mathcal{T}$, $\mathcal{L}_h^{\text{RL}}$ denotes the RL loss for $\pi_h$, and $\mathcal{B}_h^{\mathcal{T}} = \{\tau_h^{\mathcal{T}}\}$ is the high-level buffer that stores $\tau_h^{\mathcal{T}}$ for each $\mathcal{T} \in \mathcal{M}_{\text{train}}$. Here, the reward sums $\sum_{t=kH_s}^{(k+1)H_s-1} r_t$ are obtained via environment interactions of $a_t \sim \pi_l(\cdot|s_t, z_{kH_s})$ for $t = kH_s, \cdots, (k+1)H_s - 1$ with $k = 0, \cdots$. During meta-test, the high-level policy is adapted using a limited number of samples, showing good performance on long-horizon tasks.

## 4    METHODOLOGY

### 4.1    MOTIVATION: TOWARD ROBUST SKILL LEARNING UNDER NOISY DEMONSTRATIONS

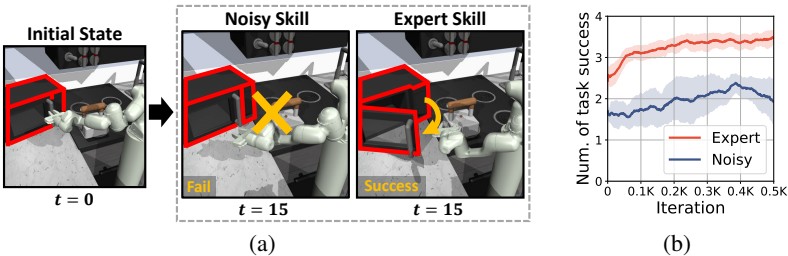

Figure 2: Comparison of prior skill learning method in microwave-opening task: (a) Learned skills with expert and noisy demonstrations. (b) Meta-RL performance with learned skills. Low-level skills are learned via SPiRL, while the meta-learning component follows the structure of SiMPL.

Most existing skill-based meta-RL approaches discussed in Section 3 assume clean offline demonstrations, but real-world datasets are often corrupted by noise from aging hardware, disturbances, or sensor drift. Unlike online training that can adapt through continuous re-training, static offline datasets are particularly susceptible to such noise. This issue becomes critical in long-horizon tasks, where errors accumulate, and in precise manipulation tasks that require reliable execution. Fig. 2(a) illustrates this problem: in the Kitchen microwave-opening task, skills learned from expert demonstrations complete the task successfully, whereas skills learned from noisy data fail even to grasp the handle. This results in a significant downstream performance drop, as shown in Fig. 2(b), where noisy skills lead to poor task success rates and unstable training curves, with each iteration denoting a training loop consisting of policy rollout and update. The root cause is that existing methods treat all trajectories equally, allowing low-quality samples to dominate skill learning.

To address this, we propose the Self-Improving Skill Learning (SISL) framework, which enhances meta-RL by introducing a decoupled skill improvement policy. The high-level policy maximizes returns using the current skill library, while the improvement policy independently perturbs trajectories near the offline data distribution to discover higher-quality variants. The resulting trajectories are selectively stored and prioritized, then used to refine the skill encoder and low-level policy. Through this iterative refinement, SISL progressively denoises the skill library and improves generalization. We further incorporate prioritized buffering and maximum return relabeling as auxiliary mechanisms that enhance sample efficiency and accelerate convergence.

### 4.2    SELF-IMPROVING SKILL REFINEMENT WITH DECOUPLED POLICIES

We now describe the proposed Self-Improving Skill Learning (SISL) framework, which formalizes the iterative refinement process motivated in Section 4.1. We adopt the standard skill-based meta-RL setup where the skill encoder $q(z|\tilde{\tau}_{t:t+H_s})$ extracts a latent skill $z$ from trajectory segments, the high-level policy $\pi_h(z|s, e^{\mathcal{T}})$ selects $z$ every $H_s$ steps, and the low-level skill policy $\pi_l(a|s, z)$ executes the chosen skill for $H_s$ steps. Building on this setup, SISL decouples training into two complementary components: *a high-level policy $\pi_h$* that exploits the current skill library to maximize return, and *a skill-improvement policy $\pi_{\text{imp}}(a_t|s_t, i)$*, defined for each training task $\mathcal{T}_i$ $(i = 1, \ldots, N_{\mathcal{T},\text{train}})$, that deliberately perturbs trajectories near the offline data distribution to discover improved behaviors. This decoupling enables simultaneous exploitation and targeted skill

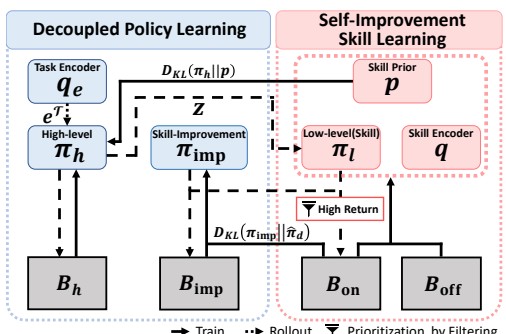
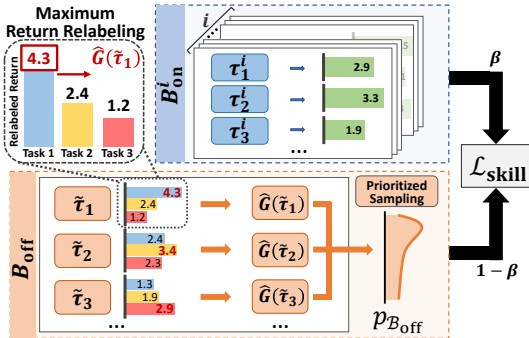

Figure 3: Proposed SISL framework. The decoupled policy learning phase uses ($\pi_h$, $\pi_{\text{imp}}$) to solve tasks and discover improved behaviors. The skill learning phase periodically updates the skill components ($\pi_l$, $p$, $q$).

Figure 4: Maximum return relabeling. After relabeling to the online buffers $\mathcal{B}_{\text{on}}^i$ and the offline buffer $\mathcal{B}_{\text{off}}$ using the learned reward model, SISL performs prioritized skill refinement by mixing both buffers via coefficient $\beta$.

improvement. Perturbations are guided by intrinsic motivation signals (Burda et al., 2018) to encourage coverage of diverse trajectories, although any novelty-driven exploration mechanism could be used. Importantly, unlike generic novelty-driven exploration, $\pi_{\text{imp}}$ restricts perturbations to remain close to the offline data manifold, producing realistic trajectories that can be effectively used to refine $\pi_l$. However, when the offline dataset contains noise, training $\pi_{\text{imp}}$ near this distribution in the early stages may be hindered by low-quality samples, reducing the effectiveness of skill improvement.

To overcome this issue, SISL employs a self-supervised guidance mechanism using two additional buffers: The improvement buffer $\mathcal{B}_{\text{imp}}^i = \{\tau_{\text{imp}}^i\}$ stores all trajectories generated by $\pi_{\text{imp}}$, where each trajectory is $\tau_{\text{imp}}^i = (s_0^i, a_0^i, r_0^i, \ldots, s_H^i)$ with $a_t^i \sim \pi_{\text{imp}}(\cdot|s_t, i)$, and is directly used to update $\pi_{\text{imp}}$ itself, and the prioritized online buffer $\mathcal{B}_{\text{on}}^i = \{\tau_{\text{high}}^i\}$ selectively retains the highest-return trajectories, where each $\tau_{\text{high}}^i$ is chosen based on its return $G(\tau^i) = \sum_{t=0}^{H-1} \gamma^t r_t^i$. This buffer is initialized with the offline dataset $\mathcal{B}_{\text{off}}$ and gradually becomes dominated by improved trajectories generated by both $\pi_{\text{imp}}$ and $\pi_h$. This buffer serves two purposes: (1) it provides a cleaner, progressively improving dataset that supervises $\pi_{\text{imp}}$ in a self-supervised manner, guiding it toward regions that have empirically led to success, and (2) it supplies high-quality samples for refining the skill encoder $q$, skill prior $p$, and low-level policy $\pi_l$.

The skill-improvement policy is then trained so that its trajectory distribution increasingly matches the successful samples in $\mathcal{B}_{\text{on}}^i$ while still exploring variations through controlled perturbations:

$$\sum_i \mathbb{E}_{\tau^i \sim \mathcal{B}_{\text{imp}}^i \cup \mathcal{B}_{\text{on}}^i} \left[ \mathcal{L}_{\text{imp}}^{\text{RL}}(\pi_{\text{imp}}) \right] + \lambda_{\text{imp}}^{\text{kld}} \mathbb{E}_{\tau^i \sim \mathcal{B}_{\text{on}}^i} \mathcal{D}_{\text{KL}}(\hat{\pi}_d^i \| \pi_{\text{imp}}), \tag{3}$$

where $\lambda_{\text{imp}}^{\text{kld}}$ is the KLD coefficient and $\hat{\pi}_d^i$ denotes the empirical action distribution derived from $\mathcal{B}_{\text{on}}^i$. $\mathcal{L}_{\text{imp}}^{\text{RL}}$ consists of the standard RL loss combined with intrinsic reward terms for perturbation, with details provided in Appendix B. This loss encourages $\pi_{\text{imp}}$ to iteratively focus on more promising state-action regions discovered so far, effectively turning $\mathcal{B}_{\text{on}}^i$ into a self-improving curriculum that steers exploration away from noisy or uninformative samples.

Building on the improved trajectory distribution obtained from this update, SISL performs skill refinement in dedicated phases every $K_{\text{iter}}$ iterations rather than after every update step. At the end of each phase, the skill encoder $q$, skill prior $p$, and low-level policy $\pi_l$ are re-trained on both $\mathcal{B}_{\text{off}}$ and $\mathcal{B}_{\text{on}}^i$ using the SPiRL objective in Eq. (1), and the high-level policy $\pi_h$ is reinitialized to fully benefit from the updated skill library. This periodic refinement mitigates bias from outdated skill embeddings and accelerates adaptation, as shown in our ablation study in Appendix G.4. These phases yield progressively denoised skills and a stable training signal, enabling $\pi_h$ to solve tasks more efficiently as training progresses. The overall SISL structure is illustrated in Fig. 3. However, the large number of noisy trajectories in $\mathcal{B}_{\text{off}}$ can still reduce skill learning efficiency. The next section introduces a maximum return relabeling mechanism that prioritizes the most relevant offline trajectories so that only high-value samples significantly influence skill refinement.

### 4.3 Skill Prioritization via Maximum Return Relabeling

The proposed SISL refines the low-level skills by leveraging both the offline dataset $\mathcal{B}_{\text{off}}$ and the prioritized online buffers $\mathcal{B}_{\text{on}}^i$ for each training task $\mathcal{T}_i$. While $\mathcal{B}_{\text{on}}^i$ provides high-quality trajectories collected from the skill-improvement policy $\pi_{\text{imp}}$ and successful rollouts from $\pi_h$, relying solely on it risks overfitting to a narrow distribution and limiting generalization. Conversely, $\mathcal{B}_{\text{off}}$ provides diverse trajectories that are beneficial for generalization but also contains many noisy or suboptimal rollouts, which can degrade skill quality if sampled uniformly. To address this trade-off, we introduce skill prioritization via a *maximum return relabeling* mechanism that assigns hypothetical returns to offline trajectories and reweights samples in both $\mathcal{B}_{\text{off}}$ and $\mathcal{B}_{\text{on}}^i$ according to their estimated task relevance. Specifically, maximum return relabeling assigns each trajectory $\tilde{\tau} \in \mathcal{B}_{\text{off}}$ a hypothetical return that reflects its potential contribution to task success. To compute this return, SISL trains a reward model $\hat{R}(s_t, a_t, i)$ for each task $\mathcal{T}_i$, optimized with the regression loss

$$\mathbb{E}_{(s_t^i, a_t^i, r_t^i) \sim \mathcal{B}_{\text{imp}}^i \cup \mathcal{B}_{\text{on}}^i} \left[ (\hat{R}(s_t^i, a_t^i, i) - r_t^i)^2 \right], \tag{4}$$

where the targets $r_t^i$ come from improved trajectories generated by $\pi_{\text{imp}}$ and stored in $\mathcal{B}_{\text{on}}^i$. Since this regression is performed online using actual environment rewards, it remains stable throughout training. Using the trained reward model, SISL computes for each $\tilde{\tau}$ the maximum return

$$\hat{G}(\tilde{\tau}) := \max_i \left\{ \sum_t \gamma^t \hat{R}(s_t, a_t, i) \right\}, \tag{5}$$

which represents the highest predicted cumulative reward across all training tasks. The offline trajectories are then sampled according to a softmax distribution $P_{\mathcal{B}_{\text{off}}}(\tilde{\tau}) = \text{Softmax}(\hat{G}(\tilde{\tau})/T)$, where $T > 0$ is a temperature parameter controlling prioritization sharpness. This procedure biases sampling toward promising trajectories while suppressing noisy or irrelevant ones, resulting in a cleaner training signal for skill learning. The resulting skill learning objective becomes

$$\mathcal{L}_{\text{skill}}(\pi_l, q, p) := (1-\beta)\mathbb{E}_{\substack{\tilde{\tau} \sim P_{\mathcal{B}_{\text{off}}}, \\ z \sim q(\cdot|\tilde{\tau})}} [\mathcal{L}(\pi_l, q, p, z)] + \frac{\beta}{N_{\mathcal{T}, \text{train}}} \sum_i \mathbb{E}_{\substack{\tau^i \sim \mathcal{B}_{\text{on}}^i, \\ z \sim q(\cdot|\tau^i)}} [\mathcal{L}(\pi_l, q, p, z)], \tag{6}$$

where $\mathcal{L}(\pi_l, q, p, z)$ is the SPiRL skill loss defined in Eq. (1). Since $\mathcal{B}_{\text{on}}$ already contains high-return trajectories, samples are drawn uniformly from this buffer. In addition, the mixing coefficient $\beta$ is computed dynamically from the offline and online datasets based on their average returns, adaptively balancing their contributions during training as

$$\beta = \frac{\exp(\bar{G}_{\text{on}}/T)}{\exp(\bar{G}_{\text{on}}/T) + \exp(\bar{G}_{\text{off}}/T)}, \tag{7}$$

where $\bar{G}_{\text{off}}$ is the mean $\hat{G}$ across $\mathcal{B}_{\text{off}}$ and $\bar{G}_{\text{on}}$ is the mean return in $\mathcal{B}_{\text{on}}^i$. Fig. 4 illustrates the prioritization process, showing how $\beta$ dynamically balances contributions from offline and online datasets. This mechanism ensures the selection of task-relevant trajectories from both datasets, facilitating efficient training of the low-level policy. As a result, meta-training yields a refined low-level policy $\pi_l$ and a high-level policy $\pi_h$ optimized over progressively cleaner data. During meta-test, $\pi_l$ is frozen and $\pi_h$ is adapted to unseen tasks using a small number of interaction trajectories, following the other skill-based meta-RL methods. Additional implementation details for the meta-train and meta-test phases, along with the algorithm table, are provided in Appendix B.

## 5 Experiment

In this section, we evaluate the robustness of the SISL framework to noisy demonstrations in long-horizon environments and analyze how self-improving skill learning enhances performance.

### 5.1 Experimental Setup

We compare the proposed SISL with 3 non-meta RL baselines: **SAC**, which trains test tasks directly without using the offline dataset; **SAC+RND**, which incorporates RND-based intrinsic noise for enhanced exploration; and **SPiRL**, which learns skills from the offline dataset using Eq. (1)

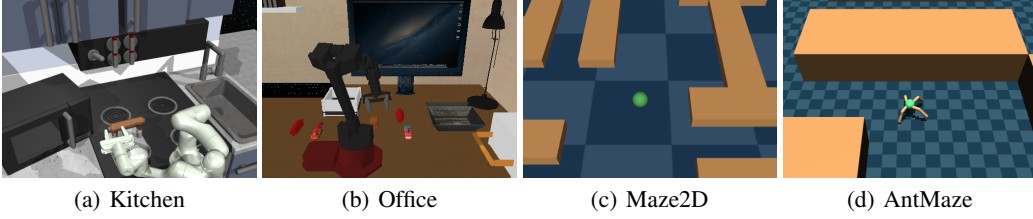

(a) Kitchen       (b) Office       (c) Maze2D       (d) AntMaze

Figure 5: Considered long-horizon environments

and trains high-level policies for individual tasks. Also, we include 4 meta-RL baselines: **PEARL**, a widely used context-based meta-RL algorithm without skill learning; **PEARL+RND**, which integrates RND-based exploration into PEARL; **SiMPL**, which applies skill-based meta-RL using Eq. (2); and our **SISL**. SISL's hyperparameters primarily follow Nam et al. (2022), with additional parameters (e.g., temperature $T$) tuned via hyperparameter search, while other baselines use author-provided code. Although SISL introduces an additional improvement policy $\pi_{\text{imp}}$, we ensured a fair comparison by keeping the total amount of training and interaction the same. Specifically, in each iteration only half of the sampled meta-train tasks use $\pi_h$ and the other half use $\pi_{\text{imp}}$, so the total number of samples and policy updates does not exceed that of SiMPL. Results are averaged over 5 random seeds, with standard deviations represented as shaded areas in graphs and $\pm$ values in tables.

## 5.2 ENVIRONMENTAL SETUP

We evaluate SISL across four long-horizon, multi-task environments: Kitchen and Maze2D from Nam et al. (2022), and Office and AntMaze, newly introduced in this work, as illustrated in Fig. 5. Offline datasets $\mathcal{B}_{\text{off}}$ are generated by perturbing expert policies with varying levels of Gaussian action noise, tailored to each environment. In the Kitchen environment, based on the Franka Kitchen from the D4RL benchmark (Fu et al., 2020b) and proposed by Gupta et al. (2020), a robotic arm completes a sequence of subtasks, with noise levels ranging from expert to $\sigma = 0.1$, $0.2$, and $0.3$. The Office environment, adapted from Pertsch et al. (2022), involves picking and placing randomly selected objects into containers, with Gaussian noise applied at the same levels as Kitchen. Maze2D, based on D4RL (Fu et al., 2020b), requires a point-mass agent to navigate a large 20x20 maze, while AntMaze features a more complex ant agent maneuvering through a 10x10 maze. In both Maze2D and AntMaze, Gaussian noise is introduced at higher levels of $\sigma = 0.5$, $1.0$, and $1.5$. Each environment is structured with distinct meta-train and meta-test tasks, ensuring that test tasks involve different goals from those seen during training. Further experimental details, including descriptions of other baselines, environment configurations (number of tasks, state representations, and reward setups), initial data collection processes, and hyperparameter settings, are provided in Appendix C.

## 5.3 PERFORMANCE COMPARISON

We compare SISL with various baseline algorithms. Non-meta RL algorithms are trained directly on each test task for 0.5K iterations due to the absence of a meta-train phase. Meta-RL algorithms undergo meta-training for 10K iterations in Kitchen and Office, and 4K in Maze2D and AntMaze, followed by fine-tuning on test tasks for an additional 0.5K iterations. For SISL, the skill refinement interval $K_{\text{iter}}$ is set to 2K for Kitchen, Office, and AntMaze; 1K for Maze2D. To ensure a fair comparison, SISL counts each update process from its skill-improvement policy and high-level policy as one iteration. Table 1 shows final average return across test tasks after the specified test iterations. In these environments, Kitchen and Office assign a reward of 1 for each successfully completed subtask, while Maze2D and AntMaze give a reward of 1 upon reaching the goal, meaning the return directly corresponds to the success rate. The corresponding learning curves are in Appendix D.1 for a more detailed comparison. From the result, SAC and PEARL baselines, which do not utilize skills or offline datasets, perform poorly on long-horizon tasks, yielding a single result across all noise levels. In contrast, SPiRL, SiMPL, and SISL, which leverage skills, achieve better performance.

SPiRL and SiMPL, however, show sharp performance declines as dataset noise increases. While both perform well with expert data, SiMPL struggles under noisy conditions due to instability in its task encoder $q_e$, sometimes performing worse than SPiRL. Here, the baseline results for Maze2D (Expert) are somewhat lower than those reported in the SiMPL paper. This discrepancy likely arises because, in constructing the offline dataset, we considered fewer tasks compared to SiMPL, result-

Table 1: Performance comparison: Final test average return for all considered environments

| Environment(Noise) | SAC | SAC+RND | PEARL | PEARL+RND | SPiRL | SiMPL | SISL |
|---|---|---|---|---|---|---|---|
| Kitchen(Expert) | | | | | 3.11±0.33 | 3.40±0.18 | **3.97**±0.09 |
| Kitchen($\sigma = 0.1$) | 0.01±0.01 | 0.02 ±0.05 | 0.23 ±0.14 | 0.42±0.16 | 3.37±0.31 | 3.76±0.14 | **3.91**±0.12 |
| Kitchen($\sigma = 0.2$) | | | | | 2.06±0.43 | 2.18±0.33 | **3.73**±0.16 |
| Kitchen($\sigma = 0.3$) | | | | | 0.83±0.17 | 0.81±0.25 | **3.48**±0.07 |
| Office(Expert) | | | | | 0.65±0.24 | 2.50±0.26 | **2.86**±0.35 |
| Office($\sigma = 0.1$) | 0.00±0.00 | 0.00±0.00 | 0.01±0.01 | 0.01±0.01 | 0.91±0.31 | 3.33±0.39 | **3.40**±0.38 |
| Office($\sigma = 0.2$) | | | | | 0.49±0.22 | 1.20±0.24 | **2.01**±0.24 |
| Office($\sigma = 0.3$) | | | | | 0.42±0.14 | 0.11±0.04 | **1.68**±0.15 |
| Maze2D(Expert) | | | | | 0.77±0.06 | 0.80±0.04 | **0.87**±0.05 |
| Maze2D($\sigma = 0.5$) | 0.20±0.06 | 0.35±0.07 | 0.10±0.01 | 0.11±0.08 | **0.89**±0.03 | 0.87±0.05 | **0.89**±0.03 |
| Maze2D($\sigma = 1.0$) | | | | | 0.80±0.01 | 0.87±0.05 | **0.93**±0.05 |
| Maze2D($\sigma = 1.5$) | | | | | 0.81±0.05 | 0.68±0.06 | **0.99**±0.02 |
| AntMaze(Expert) | | | | | 0.64±0.09 | 0.67±0.07 | **0.81**±0.08 |
| AntMaze($\sigma = 0.5$) | 0.00±0.00 | 0.00±0.00 | 0.00±0.00 | 0.00±0.00 | 0.76±0.10 | 0.77±0.05 | **0.82**±0.05 |
| AntMaze($\sigma = 1.0$) | | | | | 0.50±0.06 | 0.33±0.09 | **0.60**±0.02 |
| AntMaze($\sigma = 1.5$) | | | | | 0.30±0.01 | 0.27±0.05 | **0.41**±0.01 |

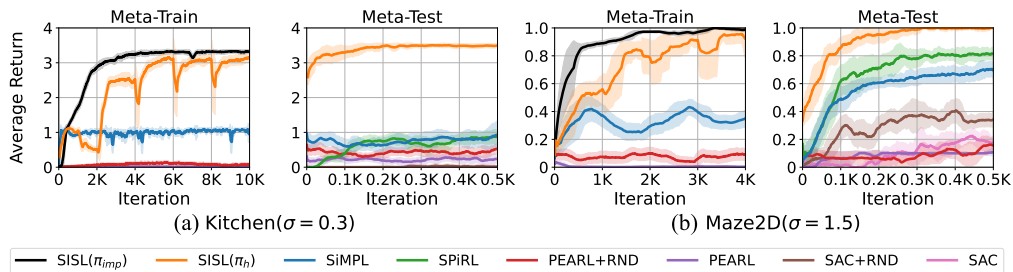

Figure 6: Learning curves of the meta-train and meta-test phases on Kitchen ($\sigma = 0.3$) and Maze2D ($\sigma = 1.5$). SISL ($\pi_{\mathrm{imp}}$) and SISL ($\pi_h$) denote the performance of the skill-improvement policy $\pi_{\mathrm{imp}}$ and high-level policy $\pi_h$ during meta-training.

ing in trajectories that do not fully cover the map. Interestingly, minor noise occasionally boosts performance by introducing diverse trajectories that improve skill learning, a detailed analysis of changes in state coverage is provided in Appendix D.1. In contrast, SISL demonstrates superior robustness across all evaluated environments, consistently outperforming baselines at varying noise levels. For example, in the Kitchen environment, SISL maintains strong performance under significant noise by effectively refining useful skills, while in Maze2D, higher noise levels lead to the improvement of diverse skills, achieving perfect task completion when $\sigma = 1.5$. These results highlight SISL's ability to discover improved behavior and refine robust skills, significantly enhancing meta-RL performance. Moreover, SISL excels with both noisy and expert data, achieving superior test performance by learning more effective skills.

Fig. 6 shows the learning progress during the meta-train/test phases for Kitchen ($\sigma = 0.3$) and Maze2D ($\sigma = 1.5$), highlighting the performance gap between SISL and other methods. The periodic drops in SISL's high-level performance correspond to the reinitialization of $\pi_h$ every $K_{\mathrm{iter}}$. Non-meta RL algorithms, including those with RND-based exploration, struggle with long-horizon tasks, while SPiRL and SiMPL show limited improvement due to their reliance on noisy offline datasets. In contrast, SISL's self-improving skill refinement supports continuous skill improvement, resulting in superior meta-test performance. To further assess SISL's robustness, we perform experiments with limited offline data, random noise injection, and diverse sub-optimal datasets in Appendix D, reflecting real-world challenges such as costly data collection and unstructured anomalies. Even under these conditions, SISL consistently outperforms the baselines, demonstrating strong robustness. Furthermore, we provide a computational complexity comparison of SISL, its ablation variants, and baseline methods in Appendix E. The results show that SISL requires only about 16% more computation time per iteration during meta-training, while the meta-test cost remains unchanged compared to SiMPL. Although SISL introduces the improvement policy $\pi_{\mathrm{imp}}$, the overall training cost remains similar because the total number of samples and updates matches SiMPL by splitting training tasks evenly between $\pi_{\mathrm{imp}}$ and $\pi_h$ for fair comparison, as described in Section 5.1. The remaining 16% overhead comes primarily from the skill refinement and reward model training. Notably, even with extended training, SiMPL fails to achieve further performance gains, highlighting SISL's advantage.

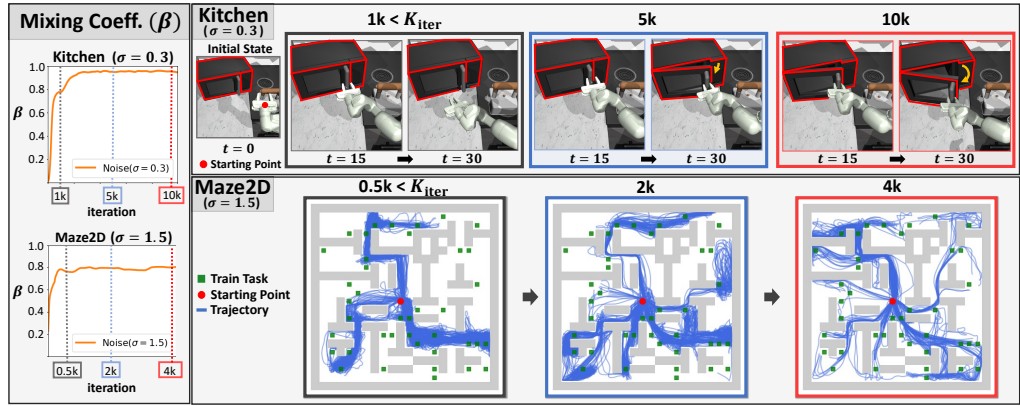

Figure 7: Visualization of buffer mixing coefficient $\beta$ dynamics and refined skill evolution in Kitchen ($\sigma = 0.3$) and Maze2D ($\sigma = 1.5$). In Kitchen, the refined skills at $t = 15$ and $t = 30$ during the microwave-opening task are depicted, while in Maze2D, trajectories using refined skills illustrate the process of progressively expanding to broader areas to solve tasks.

## 5.4 IN-DEPTH ANALYSIS OF THE PROPOSED SKILL REFINEMENT PROCESS

To analyze skill refinement and prioritization in more detail, Fig. 7 illustrates the evolution of the buffer mixing coefficient $\beta$ and skill refinement in Kitchen ($\sigma = 0.3$) and Maze2D ($\sigma = 1.5$). For Kitchen, the microwave-opening subtask is evaluated, while Maze2D focuses on navigation improvements. In the early stages (1K iterations for Kitchen, 0.5K for Maze2D), pretrained skills from the offline dataset are used without updates, resulting in poor performance, with the agent failing to grasp the handle in Kitchen and producing noisy trajectories in Maze2D. As training progresses, $\beta$ increases to shift contribution from offline data to newly collected high-quality data, providing task-relevant refinement. By design, the increase of $\beta$ tracks task-return improvement and serves as a soft curriculum that prevents abrupt distribution shift. At the same time, prioritized offline samples keep $\beta$ below 1, thereby ensuring generalization.

As a result, by iteration 5K in Kitchen, the agent learns to open the microwave, refining this skill to complete the task more efficiently by iteration 10K. In Maze2D, the agent explores more diverse trajectories over iterations, ultimately solving all training tasks by iteration 4K. These results highlight how SISL refines skills iteratively by leveraging prioritized data from offline and online buffers. As shown in Table 1, the learned skills generalize effectively to unseen test tasks, demonstrating SISL's robustness and efficacy. While variations in return scales across tasks can introduce bias in reward model training, our experimental environments adopt a simple reward structure based on subtask completion, under which the reward model remains stable, as shown in Appendix F.4. For environments with more complex reward functions, per-task reward standardization can be considered. In skill-based settings, both the offline trajectories and the online training tasks share similar underlying subtasks, such as region-to-region transitions in Ant/Point environments or object-centric subtasks in Kitchen/Office domains. Because these subtasks define reward semantics consistently across training tasks and offline dataset, this relabeling process does not introduce additional instability. In addition, our softmax-based prioritization forms a distribution rather than relying on a single trajectory, which further mitigates the impact of any minor estimation error. We provide $\beta$ trends, compare refined skills via skill trajectory visualization, task-representation improvements, and policy skill composition in Appendix F. These analyses provide insights into SISL's effectiveness in optimizing skill execution and enhancing task representation.

## 5.5 ABLATION STUDIES

We evaluate the impact of SISL's components and key hyperparameters in Kitchen ($\sigma = 0.3$) and Maze2D ($\sigma = 1.5$), focusing on the effect of the prioritization temperature $T$.

**Component Evaluation:** To evaluate the importance of SISL's components, we compare the meta-test performance of SISL with all components included against the following variations: (1) Without $\mathcal{B}_{\text{off}}$, relying solely on $\mathcal{B}_{\text{on}}^i$, assessing the influence of offline data in skill refinement; (2) Without

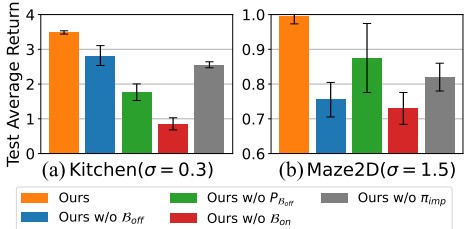

Figure 8: Component evaluation

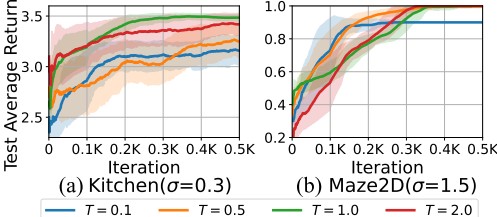

Figure 9: Impact of prioritization temperature $T$

$P_{\mathcal{B}_{\text{off}}}$, applying uniform sampling in $\mathcal{B}_{\text{off}}$ instead of maximum return relabeling, to test the effect of prioritization; (3) Without $\mathcal{B}_{\text{on}}$, using only $\mathcal{B}_{\text{off}}$, examining the contribution of high-quality samples obtained by $\pi_{\text{imp}}$ and $\pi_h$; and (4) Without $\pi_{\text{imp}}$, removing the skill-improvement policy, verifying whether exploration near the skill distribution discovers improved behaviors. Fig. 8 presents the comparison results, demonstrating significant performance drops when either buffer is removed, highlighting their critical role in effective skill discovery. Uniform sampling in $\mathcal{B}_{\text{off}}$ also reduces performance, underlining the importance of maximum return relabeling. Lastly, excluding $\pi_{\text{imp}}$ notably degrades performance, emphasizing the critical role of discovering improved behavior.

**Prioritization Temperature $T$:** The prioritization temperature $T$ adjusts the prioritization between online and offline buffers. Specifically, lower $T$ biases sampling toward high-return buffers, while higher $T$ results in uniform sampling. Fig. 9 illustrates the performance variations with different prioritization temperatures $T$. When $T = 0.1$, performance degrades due to excessive focus on a single buffer, aligning with the trends observed in the component evaluation. Conversely, high $T = 2.0$ also degrades performance by eliminating prioritization. These results highlight the importance of proper tuning: $T = 1.0$ for Kitchen and $T = 0.5$ for Maze2D achieve the best performance. Based on the result, we set $T$ approximately proportional to the high-return range of each environment, which consistently yielded the best performance while avoiding extensive tuning.

To provide a clearer understanding of SISL's components, we include several additional ablations in the Appendix. First, we analyze the temperature $T$ for all environments and noise levels in Appendix G.2, showing that its optimal value depends mainly on the return scale of each environment. Second, we study the KLD coefficient $\lambda_{\text{imp}}^{\text{kld}}$ in Appendix G.3 and find that performance is stable across a wide range of values, but drops sharply when the coefficient is zero because the guiding effect of the KL term disappears. Third, we examine the reinitialization interval $K_{\text{iter}}$ in Appendix G.5, confirming that the high-level policy needs a minimum amount of adaptation time after each refinement step, while larger intervals produce similar outcomes as long as skills are updated a few times during training. Together, these results highlight the role and influence of each component in SISL.

# 6 LIMITATION

Although SISL demonstrates strong performance, it has several limitations. First, although SISL achieves notable performance gains, its computation time per iteration increases by 16% over the baseline. This overhead mainly comes from training the skill model without freezing it during meta-training, but the performance table and ablation study confirm that this component is essential. (See Appendix E for details) Second, SISL requires fine-tuning during the meta-test phase for optimal performance, which introduces additional computational overhead. Addressing this through zero-shot skill adaptation could enhance its practicality, enabling transfer to new tasks without retraining. Future work in this direction could significantly improve SISL's applicability in real-world scenarios.

# 7 CONCLUSION

In this paper, we propose SISL, a robust skill-based meta-RL framework designed to address noisy offline demonstrations in long-horizon tasks. Through self-improving skill refinement and prioritization via maximum return relabeling, SISL effectively prioritizes task-relevant trajectories for skill learning and enables efficient exploration and targeted skill optimization. Experimental results highlight its robustness to noise and superior performance across various environments, demonstrating its potential for scalable meta-RL in real-world applications where data quality is critical.

## ACKNOWLEDGMENT

This work was supported partly by the Institute of Information & Communications Technology Planning & Evaluation (IITP) grant funded by the Korea government (MSIT) (No. RS-2022-II220469, Development of Core Technologies for Task-oriented Reinforcement Learning for Commercialization of Autonomous Drones), (No. RS-2025-25442824, AI Star-Fellowship Program (UNIST)), and (No. RS-2020-II201336, Artificial Intelligence Graduate School Support (UNIST)), and partly by the National Research Foundation of Korea (NRF) grant funded by the Korea government (MSIT) (No. RS-2025-23523191, LLM-Based Multi-Agent Reinforcement Learning for End-to-End Large Autonomous Swarm Control).

## ETHICS STATEMENT

By introducing self-improving skill refinement and skill prioritization via maximum return relabeling, SISL improves the stability and generalizability of skill learning, allowing agents to adapt rapidly to new tasks even when data quality is imperfect. This advancement has the potential to make reinforcement learning more practical and reliable in real-world settings where collecting high-quality data is difficult or expensive. While this framework may have potential societal implications by enhancing reinforcement learning's real-world applicability, we believe it is primarily foundational in nature and does not introduce any new risks of malicious use.

## REPRODUCIBILITY STATEMENT

To reproduce SISL, we provide the loss function redefined as neural network parameters, along with the meta-train and meta-test algorithm tables in Appendix B. For the algorithm implementation, we provide the system specifications used for experiments, the source of the baseline algorithm, environment details, the offline dataset construction method, and the hyperparameter setup in Appendix C. Additionally, we provide the anonymized code for SISL in the supplementary material, enabling the reproduction of the proposed algorithm and experiment results.

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

## A  The use of Large Language Models (LLMs)

We wrote the entire manuscript ourselves, including the main text and the appendix. We used large language models only for copy editing to improve spelling and readability, and we verified all suggested revisions before incorporation. LLMs were not used to generate ideas, methods, analyses, results, code, figures, or citations beyond minor edits. All technical content and experiments were conceived, implemented, and validated by the authors. We manually audited every citation and numerical claim and accept full responsibility for the manuscript.

## B  Implementation Details on SISL

This section provides a detailed implementation of the proposed SISL framework. As outlined in Section 4, SISL begins with an initial skill learning phase (pre-train) to train the low-level skill policy, skill encoder, and skill prior. It then progresses to the meta-train phase, where **decoupled policy learning** is performed using high-level policy, task encoder, and skill-improvement policy. Also, **self-improvement skill learning** is executed via maximum return relabeling using reward model, low-level skill policy, skill encoder, and skill prior. Finally, in the meta-test phase, rapid adaptation to the target task is achieved via fine-tuning based on the trained high-level policy and task encoder. Section B.1 details the initial skill learning phase, Section B.2 elaborates on the meta-train phase, and Section B.3 explains the meta-test phase. All loss functions in SISL are redefined in terms of the neural network parameters of its policies and models. Additionally, the overall structure for the meta-train and meta-test phases is provided in Algorithms 1 and 2.

### B.1  Initial Skill Learning Phase

Following SPiRL (Pertsch et al., 2021), introduced in Section 3, we train initial skills using the offline dataset $\mathcal{B}_{\text{off}}$. The low-level skill policy $\pi_{l,\phi}$, skill encoder $q_\phi$, and skill prior $p_\phi$ are parameterized by $\phi$ and trained using the following loss function (modified from Eq. (1)):

$$\mathcal{L}_{\text{spirl}}(\phi)$$

$$:= \mathbb{E}_{\substack{(s_{t:t+H_s}, a_{t:t+H_s}) \sim \mathcal{B}_{\text{off}} \\ z \sim q_\phi(\cdot | s_{t:t+H_s}, a_{t:t+H_s})}} \left[ \mathcal{L}\big(\pi_{l,\phi}, q_\phi, p_\phi, z\big) \right]$$

$$= \mathbb{E}_{\substack{(s_{t:t+H_s}, a_{t:t+H_s}) \sim \mathcal{B}_{\text{off}} \\ z \sim q_\phi(\cdot | s_{t:t+H_s}, a_{t:t+H_s})}} \left[ -\sum_{k=t}^{t+H_s-1} \log \pi_{l,\phi}(a_k | s_k, z) + \lambda_l^{\text{kld}} \mathcal{D}_{\text{KL}}\Big(q_\phi(\cdot | s_{t:t+H_s}, a_{t:t+H_s}) \,\Big|\Big|\, \mathcal{N}(\mathbf{0}, \mathbf{I})\Big) \right.$$

$$\left. + \mathcal{D}_{\text{KL}}\Big(\lfloor q_\phi(\cdot | s_{t:t+H_s}, a_{t:t+H_s}) \rfloor \,\Big|\Big|\, p_\phi(\cdot | s_t)\Big) \right],$$

(B.1)

where $\lfloor \cdot \rfloor$ represents the stop gradient operator, which prevents the KL term for skill prior learning from influencing the skill encoder. Using the pre-trained $\pi_{l,\phi}$, $q_\phi$, and $p_\phi$, SISL refines skills during the meta-train phase to further enhance task-solving capabilities.

### B.2  Meta-Train Phase

As described in Section 4, SISL comprises two main processes: **Decoupled Policy Learning**, which explores task-relevant behavior near skill distribution using skill-improvement policy, and trains the high-level policy, task encoder to effectively utilize learned skills for solving tasks; and **Self-Improvement Skill Learning**, which improves skills using prioritization via maximum return relabeling. Detailed explanations for each process are provided below.

**Decoupled Policy Learning**

As described in Section 4.2, the skill-improvement policy $\pi_{\text{imp},\psi}$, parameterized by $\psi$, is designed to expand the skill distribution and discover task-relevant behaviors near trajectories stored in the prioritized on-policy buffer $\mathcal{B}_{\text{on}}^i$ for each training task $\mathcal{T}^i$. This buffer prioritizes trajectories that best solve the tasks. Additionally, the state-action value function $Q_{\text{imp},\psi}$, also parameterized by $\psi$, is defined to train the skill-improvement policy using soft actor-critic (SAC). To enhance exploration, both extrinsic reward $r_t^i$ and intrinsic reward $r_{\text{int},t}^i$ are employed. The intrinsic reward, based on

random network distillation (RND), is computed as the L2 loss between a randomly initialized target network $\hat{f}^i_{\bar{\eta}}$ and a prediction network $f^i_{\eta}$, parameterized by $\eta$ and $\bar{\eta}$, respectively, and is expressed as:

$$r^i_{\text{int},t} := \left\| f^i_{\eta}(s_{t+1}) - \hat{f}^i_{\bar{\eta}}(s_{t+1}) \right\|^2_2, \tag{B.2}$$

where $i$ is the task index, and $f_{\eta}$ is updated to minimize this loss. A dropout layer is applied to $f_{\eta}$ to prevent over-sensitivity to state $s$. The RL loss functions of SAC for training the skill-improvement policy $\pi_{\text{imp},\psi}$ and the state-action value function $Q_{\text{imp},\psi}$ using the intrinsic reward $r_{\text{int},t}$ are defined as follows:

$$\mathcal{L}^{\text{critic}}_{\text{imp}}(\psi) := \sum_i \mathbb{E}_{\substack{(s_t,a_t,r^i_t,s_{t+1})\sim\mathcal{B}^i_{\text{imp}}\cup\mathcal{B}^i_{\text{on}} \\ a_{t+1}\sim\pi_{\text{imp},\psi}(\cdot|s_t,i)}} \left[ \frac{1}{2}\bigg( Q_{\text{imp},\psi}(s_t,a_t,i) - \Big(\delta_{\text{ext}}r^i_t + \delta_{\text{int}}r^i_{\text{int,t}} + \gamma_{\text{imp}}\big(Q_{\text{imp},\psi}(s_{t+1},a_{t+1},i) \right.$$
$$\left. + \lambda^{\text{ent}}_{\text{imp}}\log\pi_{\text{imp},\psi}(a_{t+1}|s_{t+1},i)\big)\Big)\bigg)^2 \right]$$
$$\mathcal{L}^{\text{actor}}_{\text{imp}}(\psi) := \sum_i \mathbb{E}_{\substack{s_t\sim\mathcal{B}^i_{\text{imp}}\cup\mathcal{B}^i_{\text{on}} \\ a_t\sim\pi_{\text{imp},\psi}(\cdot|s_t,i)}} \left[ \lambda^{\text{ent}}_{\text{imp}}\log\pi_{\text{imp},\psi}(a_t|s_t,i) - Q_{\text{imp},\psi}(s_t,a_t,i) \right] - \lambda^{\text{kld}}_{\text{imp}}\sum_i \mathbb{E}_{(s_t,a_t)\sim\mathcal{B}^i_{\text{on}}} \left[ \log\pi_{\text{imp},\psi}(a_t|s_t,i) \right]. \tag{B.3}$$

Here, $\delta_{\text{ext}}$ and $\delta_{\text{int}}$ are extrinsic and intrinsic reward ratios, $\gamma_{\text{imp}}$ is the discount factor, $\lambda^{\text{ent}}_{\text{imp}}$ is the exploration entropy coefficient adjusted automatically by SAC, and $\lambda^{\text{kld}}_{\text{imp}}$ is the KLD coefficient. Also, note that Eq. (B.3) provides a parameterized and detailed reformulation of Eq. (3) from Section 4.2, explicitly incorporating parameterization and loss scaling details.

To mutually update skill selection based on the refined skills, the updated and fixed low-level skill policy $\bar{\pi}_{l,\phi}$ and skill prior $\bar{q}_{\phi}$ are utilized to train the high-level policy following the SiMPL framework introduced in Section 3. The objective is to select skill representations $z$ that maximize task returns while ensuring the high-level policy remains close to the skill prior for stable and efficient learning. The high-level policy $\pi_{h,\theta}$ and value function $Q_{h,\theta}$ are parameterized by $\theta$ and trained using the soft actor-critic (SAC) framework, with the RL loss functions defined as:

$$\mathcal{L}^{\text{critic}}_h(\theta) := \mathbb{E}_{\substack{(s_t,z_t,r^h_t,s_{t+H_s-1})\sim\mathcal{B}^{\mathcal{T}}_h,e^{\mathcal{T}}\sim q_{e,\theta}(\cdot|c^{\mathcal{T}}) \\ z_{t+1}\sim\pi_{h,\theta}(\cdot|s_{t+H_s-1},e^{\mathcal{T}})}} \left[ \frac{1}{2}\bigg( Q_{h,\theta}(s_t,z_t,e^{\mathcal{T}}) - \Big(r^h_t + \gamma_h\big(Q_{h,\theta}(s_{t+H_s-1},z_{t+1},e^{\mathcal{T}}) \right.$$
$$\left. - \lambda^{\text{kld}}_h\mathcal{D}_{\text{KL}}\big(\pi_{h,\theta}(\cdot|s_{t+H_s-1},e^{\mathcal{T}})\,\big|\big|\,\bar{p}_{\phi}(\cdot|s_{t+H_s-1})\big)\big)\Big)\bigg)^2 \right]$$
$$\mathcal{L}^{\text{actor}}_h(\theta) := \mathbb{E}_{\substack{s_t\sim\mathcal{B}^{\mathcal{T}}_h,e^{\mathcal{T}}\sim q_{e,\theta}(\cdot|c^{\mathcal{T}}) \\ z_t\sim\pi_{h,\theta}(\cdot|s_t,e^{\mathcal{T}})}} \left[ \lambda^{\text{kld}}_h\mathcal{D}_{\text{KL}}\Big(\pi_{h,\theta}(\cdot|s_t,e^{\mathcal{T}})\,\big|\big|\,\bar{p}_{\phi}(\cdot|s_t)\Big) - Q_{h,\theta}(s_t,z_t,e^{\mathcal{T}}) \right], \tag{B.4}$$

where $q_{e,\theta}$ is the parameterized task encoder with parameter $\theta$, $\gamma_h$ is the high-level discount factor, and $\lambda^{\text{kld}}_h$ is the high-level KLD coefficient. The term $r^h_t = \sum^{t+H_s-1}_{k=t} r_k$ represents the cumulative rewards, with states and rewards obtained by executing the low-level skill policy $\bar{\pi}_{l,\phi}$ using $z_t \sim \pi_{h,\theta}(\cdot|s_t)$ over $H_s$ timesteps. The context $c^{\mathcal{T}} = (s_k,z_k,r^h_k,s_{k+H_s-1})^{N_{\text{prior}}}_{k=1}$, where $N_{\text{prior}}$ is the number of context transitions, denotes the high-level transition set of task $\mathcal{T}$. This context is used to select the task representation $e^{\mathcal{T}}$ from the task encoder $q_{e,\theta}$. Also, note that Eq. (B.4) is a parameterized modification of Eq. (2) from Section 3.

**Self-Improvement Skill Learning**

To extract better trajectories and learn skills that effectively solve tasks, the online buffer $\mathcal{B}^i_{\text{on}}$ selectively stores high-return trajectories collected during the meta-training phase through the execution of the low-level policy $\pi_{l,\phi}$ and the skill-improvement policy $\pi_{\text{imp},\psi}$. A trajectory $\tau^i$ is added to $\mathcal{B}^i_{\text{on}}$ if its return $G(\tau^i)$ exceeds the minimum return in the buffer, $\min_{\tau'\in\mathcal{B}^i_{\text{on}}} G(\tau')$. To refine skills, maximum return relabeling is applied using the parameterized reward model $\hat{R}_{\zeta}$ with parameter $\zeta$. The reward model is trained by minimizing the following MSE loss:

$$\mathcal{L}_{\text{reward}}(\zeta) := \mathbb{E}_{(s^i_t,a^i_t,r^i_t)\sim\mathcal{B}^i_{\text{imp}}\cup\mathcal{B}^i_{\text{on}}} \left[ \left( \hat{R}_{\zeta}(s^i_t,a^i_t,i) - r^i_t \right)^2 \right]. \tag{B.5}$$

This assigns priorities to offline trajectories $\tilde{\tau} \in \mathcal{B}_{\text{off}}$ (Eq. (5)), updated for $N_{\text{priority}}$ samples per iteration.

For skill learning, the low-level skill policy $\pi_{l,\phi}$, skill encoder $q_\phi$, and skill prior $p_\phi$ are optimized using the following loss function. This incorporates both high-return trajectories from the online buffer $\mathcal{B}_{\text{on}}^i$ and trajectories from the offline buffer $\mathcal{B}_{\text{off}}$, weighted by their importance:

$$
\begin{aligned}
\mathcal{L}_{\text{skill}}(\phi) := (1-\beta)\mathbb{E}_{\substack{(s_{t:t+H_s}, a_{t:t+H_s}) \sim P_{\mathcal{B}_{\text{off}}} \\ z \sim q_\phi(\cdot | s_{t:t+H_s}, a_{t:t+H_s})}} \Big[ \mathcal{L}\big(\pi_{l,\phi}, q_\phi, p_\phi, z\big) \Big] \\
+ \frac{\beta}{N_{\mathcal{T},\text{train}}} \sum_i \mathbb{E}_{\substack{(s_{t:t+H_s}, a_{t:t+H_s}) \sim \mathcal{B}_{\text{on}}^i \\ z \sim q_\phi(\cdot | s_{t:t+H_s}, a_{t:t+H_s})}} \Big[ \mathcal{L}\big(\pi_{l,\phi}, q_\phi, p_\phi, z\big) \Big],
\end{aligned}
\tag{B.6}
$$

where $\beta$ is the mixing coefficient defined in Eq. (7), and $\mathcal{L}(\pi_{l,\phi}, q_\phi, p_\phi, z)$ is the skill learning objective defined in Eq. (B.1) for optimizing $\pi_{l,\phi}$, $q_\phi$, and $p_\phi$. During training, we update the low-level policy $\bar{\pi}_{l,\phi}$, skill encoder $\bar{q}_\phi$, and skill prior $\bar{p}_\phi$ used for the skill-based meta-RL every $K_{\text{iter}}$ iterations. Specifically, the updates are performed as follows: $\bar{\pi}_{l,\phi} \leftarrow \pi_{l,\phi}$, $\bar{q}_\phi \leftarrow q_\phi$, and $\bar{p}_\phi \leftarrow p_\phi$.

After the meta-train phase is completed, the final meta-train phase parameter is stored as $\theta_{\text{final}} \leftarrow \theta$ and is subsequently used during the meta-test phase.

### B.3 META-TEST PHASE

After completing the meta-train phase of SISL, the meta-test phase is performed on the test task set $\mathcal{M}_{\text{test}}$. In this phase, previously learned components, including the low-level skill policy $\bar{\pi}_{l,\phi}$, skill prior $\bar{p}_\phi$, and task encoder $q_{e,\theta_{\text{final}}}$, are kept fixed and are no longer updated. Only the high-level policy $\pi_{h,\theta}$ and high-level value function $Q_{h,\theta}$ are trained for each test task using the soft actor-critic (SAC) framework.

During meta-testing, for each test task $\mathcal{T}$, the task representation $e^{\mathcal{T}}$ is inferred from the fixed task encoder $q_{e,\theta_{\text{final}}}$. The SAC algorithm is then applied to optimize the high-level policy and value function for the specific test task, following the same loss functions as defined in Eq. (B.4) from the meta-training phase. This approach ensures efficient adaptation to unseen tasks by leveraging the fixed, pre-trained low-level skills and task representations.

---

**Algorithm 1:** SISL: Meta-Train Phase

---

**Require:** Training tasks $\mathcal{M}_{\text{train}}$, offline dataset $\mathcal{B}_{\text{off}}$, low-level policy $\pi_{l,\phi}$, skill encoder $q_\phi$, and skill prior $p_\phi$.

**Initialize:** High-level policy $\pi_{h,\theta}$, skill-improvement policy $\pi_{\text{imp},\psi}$, task encoder $q_{e,\theta}$, reward model $\hat{R}_\zeta$, and value functions $Q_{h,\theta}$, $Q_{\text{imp},\psi}$.

(Initial Skill Learning)

Update $\pi_{l,\phi}$, $q_\phi$, $p_\phi$ using Eq. (B.1) with $\phi \leftarrow \phi - \lambda_l^{\text{lr}} \cdot \nabla_\phi \mathcal{L}_{\text{spirl}}(\phi)$.

1 Fix $\bar{\pi}_{l,\phi} \leftarrow \pi_{l,\phi}$, $\bar{q}_\phi \leftarrow q_\phi$, and $\bar{p}_\phi \leftarrow p_\phi$.

2 **for** iteration $k = 1, 2, \cdots$ **do**

3      **for** task $i = 1$ **to** $N_{\mathcal{T},\text{train}}$ **do**

4          Collect high-level trajectories $\tau_h^i$ and low-level trajectories $\tau_l^i$ using $\pi_{h,\theta}$ with $\bar{\pi}_{l,\phi}$, $q_{e,\theta}$.

5          Collect skill-improvement trajectories $\tau_{\text{imp}}^i$ using $\pi_{\text{imp},\psi}$.

6          Filter high-return trajectories $\tau_{\text{high}}^i$ from $\tau_l^i$ and $\tau_{\text{imp}}^i$ s.t. $G > \min_{\tau' \in \mathcal{B}_{\text{on}}^i} G(\tau')$.

7          Store $\tau_h^i$, $\tau_{\text{imp}}^i$, and $\tau_{\text{high}}^i$ into $\mathcal{B}_h^i$, $\mathcal{B}_{\text{imp}}^i$, and $\mathcal{B}_{\text{on}}^i$.

8      Compute prioritization factors: $P_{\mathcal{B}_{\text{off}}}$ and $\beta$.

9      **for** gradient step **do**

         (Decoupled Policy Learning)

10          Update $\pi_{\text{imp},\psi}$, $Q_{\text{imp},\psi}$ using Eq. (B.3) with $\psi \leftarrow \psi - \lambda_{\text{imp}}^{\text{lr}} \cdot \nabla_\psi (\mathcal{L}_{\text{imp}}^{\text{critic}}(\psi) + \mathcal{L}_{\text{imp}}^{\text{actor}}(\psi))$.

11          Update $\pi_{h,\theta}$, $Q_{h,\theta}$, $q_{e,\theta}$ using Eq. (B.4) with $\theta \leftarrow \theta - \lambda_h^{\text{lr}} \cdot \nabla_\theta (\mathcal{L}_h^{\text{critic}}(\theta) + \mathcal{L}_h^{\text{actor}}(\theta))$.

         (Self-Improvement Skill Learning)

12          Update reward model $\hat{R}_\zeta$ using Eq. (B.5) with $\zeta \leftarrow \zeta - \lambda_{\text{reward}}^{\text{lr}} \cdot \nabla_\zeta \mathcal{L}_{\text{reward}}(\zeta)$.

13          Update $\pi_{l,\phi}$, $q_\phi$, $p_\phi$ using Eq. (B.6) with $\phi \leftarrow \phi - \lambda_l^{\text{lr}} \cdot \nabla_\phi \mathcal{L}_{\text{skill}}(\phi)$.

14      **if** $k \bmod K_{\text{iter}} = 0$ **then**

15          Update $\bar{\pi}_{l,\phi} \leftarrow \pi_{l,\phi}$, $\bar{q}_\phi \leftarrow q_\phi$, and $\bar{p}_\phi \leftarrow p_\phi$.

16          Reinitialize $\pi_{h,\theta}$.

Save the final meta-train phase parameter $\theta_{\text{final}} \leftarrow \theta$.

---

**Algorithm 2:** SISL: Meta-test phase

---

**Require:** Target task $\mathcal{T}$, high-level policy $\pi_{h,\theta}$, value function $Q_{h,\theta}$, task encoder $q_{e,\theta_{\text{final}}}$, low-level policy $\bar{\pi}_{l,\phi}$, and skill prior $\bar{p}_\phi$.

1 Collect context $c^{\mathcal{T}}$ using $\pi_{h,\theta}$ with $\bar{\pi}_{l,\phi}$, $e \sim \mathcal{N}(0, I)$.

2 Compute task representation $e^{\mathcal{T}} \sim q_{e,\theta_{\text{final}}}(\cdot|c^{\mathcal{T}})$.

3 **for** iteration $k = 1, 2, \ldots$ **do**

4      Collect high-level trajectory $\tau_h^{\mathcal{T}}$ using $\pi_{h,\theta}$ with $\bar{\pi}_{l,\phi}$, $e^{\mathcal{T}}$.

5      Store $\tau_h^{\mathcal{T}}$ into $\mathcal{B}_h^{\mathcal{T}}$.

6      **for** gradient step **do**

7          Update $\pi_{h,\theta}$, $Q_{h,\theta}$ using Eq. (B.4) with $\theta \leftarrow \theta - \lambda_h^{\text{lr}} \cdot \nabla_\theta (\mathcal{L}_h^{\text{critic}}(\theta) + \mathcal{L}_h^{\text{actor}}(\theta))$

---

# C DETAILED EXPERIMENTAL SETUP

In this section, we provide a detailed description of our experimental setup. The implementation is built on PyTorch with CUDA 11.7, running on an AMD EPYC 7313 CPU with an NVIDIA GeForce RTX 3090 GPU. SISL is implemented based on the official open-source code of SiMPL, available at https://github.com/namsan96/SiMPL. For the environment implementations, we used SiMPL's code for the Kitchen and Maze2D environments, SkiLD's open-source code for the Office environment at https://github.com/clvrai/skild, and D4RL's open-source code for AntMaze at https://github.com/Farama-Foundation/D4RL/tree/master.

The hyperparameters for low-level policy training were referenced from SPiRL (Pertsch et al., 2021). Additional details about the baseline algorithms are provided in Section C.1, while Section C.2 elaborates on the environments used for evaluation. Section C.3 explains the construction of offline datasets for varying noise levels, and Section C.4 details the network architectures and hyperparameter configurations for policies, value functions, and other models.

## C.1 OTHER BASELINES

Here are the detailed descriptions and implementation details of the algorithms used for performance comparison:

**SAC**

SAC (Haarnoja et al., 2018) is a reinforcement learning algorithm that incorporates entropy to improve exploration. Instead of a standard value function, SAC uses a soft value function that combines entropy, with the entropy coefficient adjusted automatically to maintain the target entropy. To enhance value function estimation, SAC employs double $Q$ learning, using two independent value functions. SAC learns tasks from scratch without utilizing meta-train tasks or offline datasets. For the Kitchen and Office environments, the discount factor $\gamma$ is set to $0.95$, while $\gamma = 0.99$ is used for Maze2D and AntMaze environments. We utilize the open-source code of SAC at https://github.com/denisyarats/pytorch_sac.

**SAC+RND**

SAC+RND combines SAC with random network distillation (RND) (Burda et al., 2018), an intrinsic motivation technique, to enhance exploration. Like SAC, it learns tasks from scratch without meta-train tasks or offline datasets. RL hyperparameters are shared with SAC, and RND-specific hyperparameters are set to match those in SISL. Additionally, for fair comparison, the ratio of extrinsic to intrinsic rewards is aligned with SISL. We utilize the open-source code of RND at https://github.com/openai/random-network-distillation.

**PEARL**

PEARL (Rakelly et al., 2019) is a context-based meta-RL algorithm that leverages a task encoder $q_e$ to derive task representations, which are then used to train a meta-policy. PEARL adapts its learned policy quickly to unseen tasks without utilizing skills or offline datasets. Unlike the original PEARL, which does not fine-tune during the meta-test phase, we modified it to include fine-tuning on target tasks for a fair comparison. We utilize the open-source code of PEARL at https://github.com/katerakelly/oyster.

**PEARL+RND**

PEARL+RND extends PEARL by incorporating RND to enhance exploration. Like SAC+RND, the ratio of extrinsic to intrinsic rewards is set to match SISL for fair comparison.

**SPiRL**

SPiRL (Pertsch et al., 2021) is a skill-based RL algorithm that first learns a fixed low-level policy from an offline dataset and then trains a high-level policy for specific tasks. SPiRL's loss function is detailed in Section 3, and for fair comparison, loss scaling is aligned with SISL. We utilize the open-source code of SPiRL at https://github.com/clvrai/spirl.

**SiMPL**

SiMPL (Nam et al., 2022) is a skill-based meta-RL algorithm that uses both offline datasets and meta-train tasks. While it shares SISL's approach of extracting reusable skills and performing meta-train and meta-test phases, SiMPL fixes the skill model without further updates during meta-training. SiMPL's loss function is also detailed in Section 3, and SiMPL's implementation uses the same hyperparameters as SISL to ensure a fair comparison. We utilize the open-source code of SiMPL at https://github.com/namsan96/SiMPL.

## C.2 ENVIRONMENTAL DETAILS

**Kitchen**

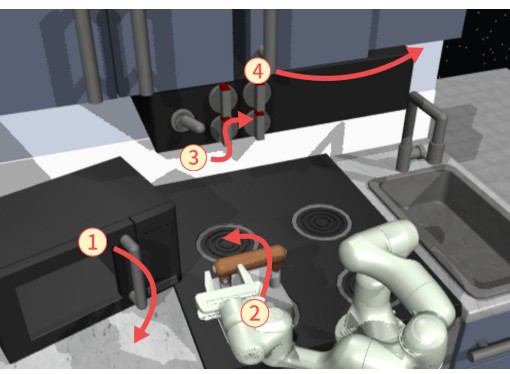

Figure C.1: Kitchen: An example of task (microwave→kettle→bottom burner→slide cabinet)

The Franka Kitchen environment is a robotic manipulation setup based on the 7-DoF Franka robot. It is introduced by Gupta et al. (2020) and later adapted by Nam et al. (2022) to exclude task information from the observation space, making it more suitable for meta-learning. The environment features seven manipulatable objects: bottom burner, top burner, light switch, slide cabinet, hinge cabinet, microwave, and kettle.

Each subtask involves manipulating one object to its target state, while a full task requires sequentially completing four subtasks. The agent earns a reward of 1 for each completed subtask, with a maximum score of 4 achievable within a 280-timestep horizon. For instance, an example task, illustrated in Fig. C.1, requires the agent to complete the sequence: microwave → kettle → bottom burner → slide cabinet. The observation space is a 60-dimensional continuous vector representing object positions and robot state information, while the action space is a 9-dimensional continuous vector. Based on the task setup from Nam et al. (2022), we expanded the meta-train task set by adding two additional tasks, resulting in a total of 25 meta-train tasks and 10 meta-test tasks. Detailed task configurations are provided in Table C.1.

Table C.1: List of meta-train tasks and meta-test tasks in Kitchen environment

| | Meta-train task | | | | | Meta-test task | | | |
|---|---|---|---|---|---|---|---|---|---|
| # | Subtask1 | Subtask2 | Subtask3 | Subtask4 | # | Subtask1 | Subtask2 | Subtask3 | Subtask4 |
| 1 | microwave | kettle | bottom burner | slide cabinet | 1 | microwave | bottom burner | light switch | top burner |
| 2 | microwave | bottom burner | top burner | slide cabinet | 2 | microwave | bottom burner | top burner | light switch |
| 3 | microwave | top burner | light switch | hinge cabinet | 3 | kettle | bottom burner | light switch | slide cabinet |
| 4 | kettle | bottom burner | light switch | hinge cabinet | 4 | microwave | kettle | top burner | hinge cabinet |
| 5 | microwave | bottom burner | hinge cabinet | top burner | 5 | kettle | bottom burner | slide cabinet | top burner |
| 6 | kettle | top burner | light switch | slide cabinet | 6 | kettle | light switch | slide cabinet | hinge cabinet |
| 7 | microwave | kettle | slide cabinet | bottom burner | 7 | kettle | bottom burner | top burner | slide cabinet |
| 8 | kettle | light switch | slide cabinet | bottom burner | 8 | microwave | bottom burner | slide cabinet | hinge cabinet |
| 9 | microwave | kettle | bottom burner | top burner | 9 | bottom burner | top burner | slide cabinet | hinge cabinet |
| 10 | microwave | kettle | slide cabinet | hinge cabinet | 10 | microwave | kettle | bottom burner | hinge cabinet |
| 11 | microwave | bottom burner | slide cabinet | top burner | | | | | |
| 12 | kettle | bottom burner | light switch | top burner | | | | | |
| 13 | microwave | kettle | top burner | light switch | | | | | |
| 14 | microwave | kettle | light switch | hinge cabinet | | | | | |
| 15 | microwave | bottom burner | light switch | slide cabinet | | | | | |
| 16 | kettle | bottom burner | top burner | light switch | | | | | |
| 17 | microwave | light switch | slide cabinet | hinge cabinet | | | | | |
| 18 | microwave | bottom burner | top burner | hinge cabinet | | | | | |
| 19 | kettle | bottom burner | slide cabinet | hinge cabinet | | | | | |
| 20 | bottom burner | top burner | slide cabinet | light switch | | | | | |
| 21 | microwave | kettle | light switch | slide cabinet | | | | | |
| 22 | kettle | bottom burner | top burner | hinge cabinet | | | | | |
| 23 | bottom burner | top burner | light switch | slide cabinet | | | | | |
| 24 | top burner | hinge cabinet | microwave | slide cabinet | | | | | |
| 25 | bottom burner | hinge cabinet | light switch | kettle | | | | | |

**Office**

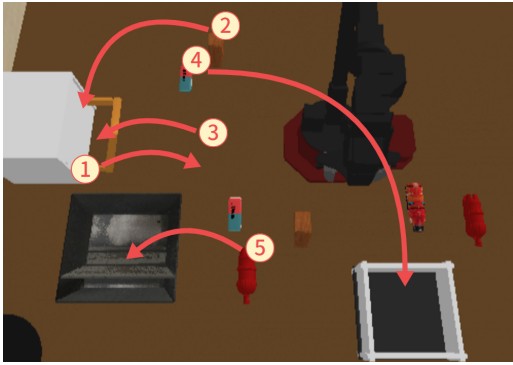

Figure C.2: Office: An example of task ((shed2, drawer)→(eraser1, container)→(pepsi2, tray))

The Office environment is a robotic manipulation setup featuring a 5-DoF robotic arm. Originally proposed by Pertsch et al. (2022), it has been modified to accommodate meta-learning tasks. The environment simulates an office cleaning scenario with seven objects (eraser1, shed1, pepsi1, gatorade, eraser2, shed2, pepsi2) and three organizers (tray, container, drawer).

The goal is to move objects to their designated organizers, with each object-to-organizer transfer constituting a subtask. A full task involves completing three sequential subtasks, where a subtask is defined as an (object, organizer) pair. For tasks involving a tray or container, the agent earns a reward of 1 for both picking and placing the object. For tasks involving the drawer, the agent receives 1 reward point for each of the following actions: opening the drawer, picking, placing, and closing the drawer. This scoring setup allows for a maximum score of 8 within a 300-timestep horizon.

An example task, depicted in Fig. C.2, requires the agent to sequentially complete: (shed2 → drawer), (eraser1 → container), and (pepsi2 → tray). The observation space is a 76-dimensional continuous vector, including object positions and robot state information, while the action space is an 8-dimensional continuous vector. The meta-train and meta-test sets include 25 and 10 tasks, respectively, similar to the configuration in the Kitchen environment. A detailed task list is provided in Table C.2.

Table C.2: List of meta-train tasks and meta-test tasks in Office environment

| | Meta-train task | | | | Meta-test task | | |
|---|---|---|---|---|---|---|---|
| # | Subtask1 | Subtask2 | Subtask3 | # | Subtask1 | Subtask2 | Subtask3 |
| 1 | (shed2, drawer) | (eraser1, container) | (pepsi2, tray) | 1 | (gatorade, drawer) | (eraser1, tray) | (pepsi2, container) |
| 2 | (shed2, container) | (eraser1, drawer) | (pepsi1, tray) | 2 | (eraser1, drawer) | (eraser2, container) | (pepsi1, tray) |
| 3 | (eraser1, tray) | (shed2, drawer) | (gatorade, container) | 3 | (eraser2, drawer) | (pepsi1, tray) | (gatorade, container) |
| 4 | (pepsi1, tray) | (eraser1, container) | (eraser2, drawer) | 4 | (shed2, drawer) | (pepsi2, tray) | (pepsi1, container) |
| 5 | (shed1, tray) | (shed2, drawer) | (pepsi2, container) | 5 | (shed2, container) | (gatorade, tray) | (eraser1, drawer) |
| 6 | (pepsi1, container) | (shed1, tray) | (eraser2, drawer) | 6 | (gatorade, container) | (eraser2, drawer) | (pepsi2, tray) |
| 7 | (gatorade, tray) | (eraser2, container) | (eraser1, drawer) | 7 | (gatorade, tray) | (shed1, container) | (eraser1, drawer) |
| 8 | (pepsi2, container) | (shed2, drawer) | (eraser1, tray) | 8 | (pepsi2, drawer) | (shed1, tray) | (pepsi1, container) |
| 9 | (shed2, drawer) | (gatorade, container) | (pepsi2, tray) | 9 | (pepsi1, tray) | (pepsi2, container) | (shed2, drawer) |
| 10 | (eraser1, container) | (pepsi2, drawer) | (shed1, tray) | 10 | (gatorade, drawer) | (pepsi1, container) | (eraser2, tray) |
| 11 | (eraser2, drawer) | (shed2, tray) | (pepsi2, container) | | | | |
| 12 | (pepsi2, container) | (shed2, drawer) | (shed1, tray) | | | | |
| 13 | (shed2, tray) | (pepsi1, container) | (eraser1, drawer) | | | | |
| 14 | (gatorade, tray) | (eraser1, drawer) | (pepsi1, container) | | | | |
| 15 | (eraser1, tray) | (shed1, drawer) | (gatorade, container) | | | | |
| 16 | (eraser2, drawer) | (gatorade, container) | (shed2, tray) | | | | |
| 17 | (shed2, tray) | (pepsi2, drawer) | (shed1, container) | | | | |
| 18 | (pepsi1, container) | (pepsi2, tray) | (eraser1, drawer) | | | | |
| 19 | (shed2, tray) | (gatorade, drawer) | (shed1, container) | | | | |
| 20 | (gatorade, tray) | (pepsi1, container) | (pepsi2, drawer) | | | | |
| 21 | (eraser1, tray) | (shed2, drawer) | (pepsi2, container) | | | | |
| 22 | (eraser1, tray) | (gatorade, drawer) | (shed2, container) | | | | |
| 23 | (pepsi1, container) | (shed2, drawer) | (eraser2, tray) | | | | |
| 24 | (gatorade, drawer) | (shed1, tray) | (pepsi2, container) | | | | |
| 25 | (eraser2, container) | (pepsi1, drawer) | (eraser1, tray) | | | | |

**Maze2D**

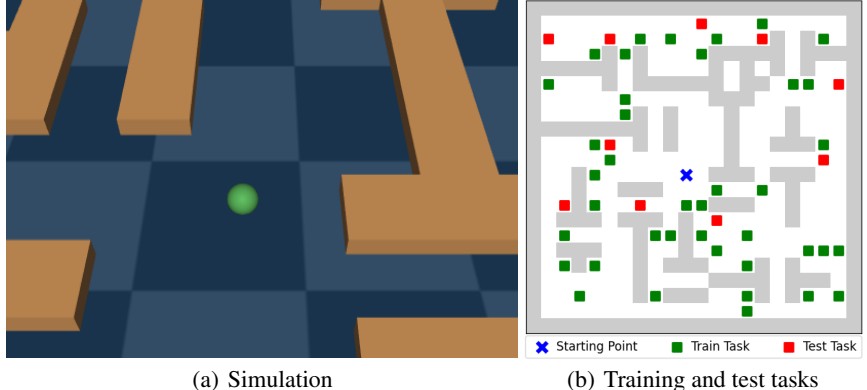

(a) Simulation      (b) Training and test tasks

Figure C.3: Maze2D: Visualization of simulation and meta-train/test tasks in Maze2D

The Maze2D environment is a navigation setup where a 2-DoF ball agent moves toward a goal point. Initially introduced by Fu et al. (2020b) and later adapted by Nam et al. (2022) for meta-learning tasks, the environment is defined on a 20x20 grid. The agent receives a reward of 1 upon reaching the goal point within a horizon of 2000 timesteps.

Fig. C.3 (a) provides a visualization of the Maze2D environment, while Fig. C.3 (b) illustrates the meta-train and meta-test tasks. In Fig. C.3 (b), green squares indicate the goal points for 40 meta-train tasks, and red squares represent the goal points for 10 meta-test tasks. All tasks share the same starting point at (10, 10), marked by a blue cross. The observation space is a 4-dimensional continuous vector containing the ball's position and velocity, while the action space is a 2-dimensional continuous vector.

**AntMaze**

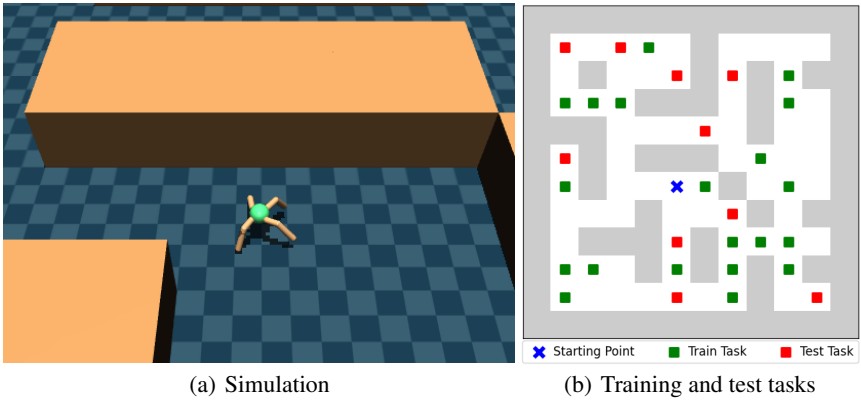

(a) Simulation      (b) Training and test tasks

Figure C.4: AntMaze: Visualization of simulation and meta-train/test tasks in AntMaze

The AntMaze environment combines navigation and locomotion, replacing the 2-DoF ball from the Maze2D environment with a more complex 8-DoF quadruped Ant robot. Initially proposed by Fu et al. (2020b) and later adapted for meta-learning setups, the environment is defined on a 10x10 grid. The agent receives a reward of 1 upon reaching the goal point within a horizon of 1000 timesteps.

Fig. C.4 (a) shows a simulation image of the AntMaze environment, and Fig. C.4 (b) depicts the meta-train and meta-test tasks. In Fig. C.4 (b), green squares mark the goal points for 20 meta-train tasks, while red squares denote the goal points for 10 meta-test tasks. All tasks share a common starting point at (5, 5), indicated by a blue cross. The observation space is a 29-dimensional continuous vector that includes the Ant's state and its $(x, y)$ coordinates, while the action space is an 8-dimensional continuous vector.

### C.3 Construction of Offline Dataset

In this section, we detail the offline datasets used in our experiments. For the Office, Maze2D, and AntMaze environments, we employ rule-based oracle controllers provided by each environment. The Office oracle controller is available at https://github.com/clvrai/skild, while the Maze2D and AntMaze oracle controllers can be found in https://github.com/Farama-Foundation/D4RL/tree/master. For the Kitchen environment, which only provides human demonstrations, we train a policy using behavior cloning to serve as the oracle controller.

For the Kitchen environment, 1M transitions are collected using 25 tasks that are not part of the training or test task sets $\mathcal{M}_{\text{train}} \cup \mathcal{M}_{\text{test}}$. Similarly, the Office environment collects 1M transitions using 80 tasks. The Maze2D and AntMaze environments follow the same approach, collecting 0.5M transitions using 40 and 50 tasks respectively, with randomly sampled initial and goal points. Unlike SiMPL, which randomly samples initial and goal points for each trajectory in the Maze2D environment, we limit our data collection to 40 distinct tasks, resulting in trajectories that do not fully cover the map. To introduce noise in the demonstrations, Gaussian noise with various standard deviations $\sigma$ is added to the oracle controller's actions. For the Kitchen and Office environments, noise levels of $\sigma = 0.1, 0.2$, and $0.3$ are used, while for Maze2D and AntMaze, $\sigma = 0.5, 1.0$, and $1.5$ are applied.

## C.4 Hyperparameter Setup

In this section, we outline the hyperparameter setup for the proposed SISL framework. For high-level policy training, we adopt the hyperparameters from SiMPL for the Kitchen and Maze2D environments. For the Office and AntMaze environments, we conduct hyperparameter sweeps using the Kitchen and Maze2D configurations as baselines.

To ensure a fair comparison, we inherit from SiMPL all hyperparameters that are shared with SISL, given its multiple loss functions, and we perform parameter sweeps only over SISL specific components, namely self-improving skill refinement and skill prioritization via maximum return relabeling. We explore prioritization temperature values $T \in [0.1, 0.5, 1.0, 2.0]$ and KLD coefficients $\lambda_{\text{imp}}^{\text{kld}} \in [0, 0.001, 0.002, 0.005]$ for skill exploration, selecting the best-performing configurations as defaults. Additionally, the ratio of intrinsic to extrinsic rewards is fixed at levels that show optimal performance in single-task SAC experiments.

For implementing the skill models ($\pi_l, q, p$), we follow SPiRL by utilizing LSTM (Graves & Graves, 2012) for the skill encoder and MLP structures for the low-level skill policy and skill prior. For implementing the high-level models ($\pi_h, Q_h, q_e$), we follow SiMPL by utilizing Set Transformer (Lee et al., 2019) for the task encoder and MLP structures for the high-level policy and value function. Additionally, for implementing the SISL, we utilize MLP structures for $\pi_{\text{imp}}$, $Q_{\text{imp}}$, and $\hat{R}$. The detailed hidden network sizes are presented in Table C.3 and Table C.4. Table C.3 presents the network architectures (the number of nodes in fully connected layers) and the hyperparameters shared across all environments, while Table C.4 details the environment-specific hyperparameter setups.

Table C.3: Network Architecture and Shared Hyperparameters

| | Group | Name | Environments | | | |
| --- | --- | --- | --- | --- | --- | --- |
| | | | Kitchen | Office | Maze2D | AntMaze |
| Shared Hyperparameters | High-level | Discount Factor $\gamma_h$ | 0.99 | | | |
| | | Learning rate $\lambda_h^{\text{lr}}$ | 0.0003 | | | |
| | | Network size $\pi_h, Q_h$ | [128]×6 | | [256]×4 | [128]×6 |
| | Low-level | Buffer size $\mathcal{B}_{\text{on}}^i$ | 10K | | | |
| | | KLD coefficient $\lambda_l^{\text{kld}}$ | 0.0005 | | | |
| | | Skill length $H_s$ | 10 | | | |
| | | Skill dimension dim($z$) | 10 | | | |
| | | # of priority update trajectory $N_{\text{priority}}$ | 200 | | | |
| | | Learning rate $\lambda_{\text{skill}}^{\text{lr}}$ | 0.001 | | | |
| | | Learning rate $\lambda_{\text{reward}}^{\text{lr}}$ | 0.0003 | | | |
| | | Network size $\hat{R}$ | [128]×3 | | | |
| | | Network size $\pi_l$ | [128]×6 | | | |
| | | Network size $p$ | [128]×7 | | | |
| | | Network size $q$ | LSTM[128] | | | |
| | Skill-Improvement | RND state dropout ratio | 0.7 | | | |
| | | RND output dimension | 10 | | | |
| | | Learning rate $\lambda_{\text{imp}}^{\text{lr}}$ | 0.0003 | | | |
| | | Network size $\pi_{\text{imp}}, Q_{\text{imp}}$ | [256]×4 | | | |
| | | Network size $f, \hat{f}$ | [128]×4 | | | |

Table C.4: Environmental Hyperparameters

| | Group | Name | Environments | | | |
| --- | --- | --- | --- | --- | --- | --- |
| | | | Kitchen | Office | Maze2D | AntMaze |
| Environmental Hyperparameters | High-level | Buffer size $\mathcal{B}_h^i$ | 3000 | 3000 | 20000 | 20000 |
| | | KLD coefficient $\lambda_h^{\text{kld}}$ | 0.03 | 0.03 | 0.001 | 0.0003 |
| | | Task latent dimension dim($e$) | 5 | 5 | 6 | 6 |
| | | Batch size (RL, per task) | 256 | 256 | 1024 | 512 |
| | | Batch size (context, per task) | 1024 | 1024 | 8192 | 4096 |
| | Low-level | Skill refinement $K_{\text{iter}}$ | 2000 | 2000 | 1000 | 2000 |
| | | Prioritization temperature $T$ | 1.0 | 1.0 | 0.5 | 0.5 |
| | Skill-Improvement | Buffer size $\mathcal{B}_{\text{imp}}^i$ | 100K | 200K | 100K | 300K |
| | | Discount factor $\gamma_{\text{imp}}$ | 0.95 | 0.95 | 0.99 | 0.99 |
| | | RND extrinsic ratio $\delta_{\text{ext}}$ | 5 | 2 | 10 | 10 |
| | | RND intrinsic ratio $\delta_{\text{int}}$ | 0.1 | 0.1 | 0.01 | 0.01 |
| | | Entropy coefficient $\lambda_{\text{imp}}^{\text{ent}}$ | 0.2 | 0.2 | 0.1 | 0.1 |
| | | KLD coefficient $\lambda_{\text{imp}}^{\text{kld}}$ | 0.005 (Expert) 0.005 ($\sigma = 0.1$) 0.002 ($\sigma = 0.2$) 0.001 ($\sigma = 0.3$) | 0.001 | 0.001 | 0.001 |

## D   ADDITIONAL COMPARISON RESULTS

In this section, we provide additional comparison results against baseline algorithms. Following Fig. 6, additional performance comparisons across all environments and noise levels are presented in Section D.1, limited offline dataset size in Section D.2, random noise injection in Section D.3, and diverse sub-optimal offline datasets in Section D.4.

### D.1   PERFORMANCE COMPARISON

Fig. D.1 presents the learning curves of average returns for the algorithms on test tasks, corresponding to the experiments summarized in Table 1. Rows represent evaluation environments, and columns denote noise levels. SISL consistently demonstrated superior robustness, outperforming all baselines across various environments and noise levels. At higher noise levels such as Noise($\sigma = 0.2$), Noise($\sigma = 0.3$) for Kitchen and Office, and Noise($\sigma = 1.0$), Noise($\sigma = 1.5$) for Maze2D and AntMaze, significant performance improvements highlight the effectiveness of skill refinement in addressing noisy demonstrations. Even with high-quality offline datasets like Expert and Noise($\sigma = 0.1$) for Kitchen and Office, and Noise($\sigma = 0.5$) for Maze2D and AntMaze, SISL further improved performance by learning task-relevant skills. These learning curves align with the trends observed in the main experiments, confirming that skill refinement enhances performance across various dataset qualities and effectively adapts to the task distribution.

Interestingly, SPiRL and SiMPL sometimes perform better with mild Gaussian noise than with expert data, especially in Maze2D and AntMaze. Our analysis suggests that mild noise increases the diversity of behaviors in the dataset without significantly reducing success rates, allowing agents to reach goals via more diverse paths. This expands state-action coverage (by about 7.3% in Maze2D and 5.2% in AntMaze), helping skill-based methods learn more flexible and reusable skills. However, as the noise level increases further, a significant portion of trajectories fail to reach the goal, leading to low quality skill learning and degraded downstream performance.

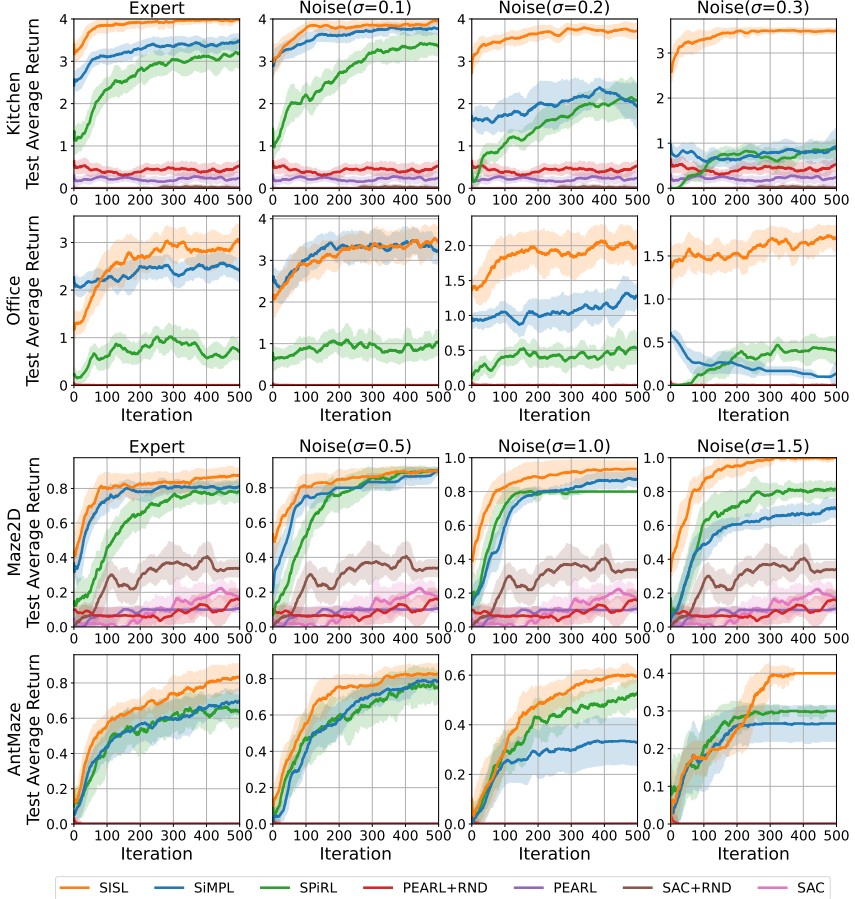

Figure D.1: Learning curves across considered environments and noise levels

## D.2 LIMITED OFFLINE DATASET SIZE

To further investigate the robustness of SISL, we conducted additional experiments to assess the impact of dataset size on skill learning and downstream performance. While SISL primarily addresses the challenge of refining corrupted skills from noisy demonstrations through online interaction, the size of the offline dataset remains an important factor, especially in practical scenarios. To assess the impact of dataset size, we conducted additional experiments on the Kitchen environment using the same expert dataset but reduced to 50% (0.5M), 25% (0.25M), and 10% (0.1M) of its original dataset size (1M). As shown in Table D.1, SISL consistently outperforms baselines even under limited expert data, and its performance degrades more gracefully compared to baselines. These results confirm SISL's ability to refine skills even from limited data, which is particularly valuable in real-world scenarios where collecting high-quality demonstrations is costly or infeasible.

Table D.1: Final performance on the Kitchen environment with varying sizes of expert datasets.

| Dataset Size | SPiRL | SiMPL | SISL |
|---|---|---|---|
| Expert(100%) | $3.11_{\pm0.33}$ | $3.40_{\pm0.18}$ | $\mathbf{3.97}_{\pm0.09}$ |
| Expert(50%) | $3.11_{\pm0.30}$ | $3.29_{\pm0.18}$ | $\mathbf{3.65}_{\pm0.06}$ |
| Expert(25%) | $2.91_{\pm0.29}$ | $2.99_{\pm0.13}$ | $\mathbf{3.60}_{\pm0.14}$ |
| Expert(10%) | $2.37_{\pm0.28}$ | $2.56_{\pm0.09}$ | $\mathbf{3.28}_{\pm0.15}$ |

## D.3 RANDOM NOISE INJECTION

While Gaussian noise is a widely used approach for degrading demonstration quality in offline RL studies, it does not fully capture the diverse and unstructured nature of real-world noise such as sensor failures, occlusions, or actuator malfunctions. In our main experiments, we adopted multi-level Gaussian noise for two reasons: (1) it is a standard and accepted method for systematically degrading demonstration quality, and (2) it enables controlled analysis of robustness under varying degrees of skill degradation. Notably, in domains such as Kitchen and Office, sufficiently high levels of Gaussian noise render learned skills nearly unusable, mimicking real-world failure scenarios in precision control tasks and providing a highly challenging regime for evaluating robustness.

To further assess the generality of our robustness claims and address concerns about the limitations of Gaussian noise, we conducted an additional experiment using random action injection in the Kitchen environment. This approach better simulates real-world anomalies such as actuator faults or sensor failures. Specifically, at each timestep, the oracle action was replaced with a randomly sampled action with a probability of 25%, 50%, or 100% (resulting in a uniformly random dataset at the highest level). This method introduces severe, unstructured corruption into the offline dataset, representing worst-case real-world failures beyond the smooth degradations induced by Gaussian noise. As shown in Table D.2, SISL consistently outperformed baseline methods, confirming its robustness even under extreme, non-gaussian corruption. These results demonstrate SISL's ability to refine useful behaviors and maintain strong performance in the presence of severe dataset corruption, further validating the generality of our robustness claims.

Table D.2: Final performance on the Kitchen environment with the random noise injection.

| Noise Type | SPiRL | SiMPL | SISL |
|---|---|---|---|
| Expert | $3.11_{\pm0.33}$ | $3.40_{\pm0.18}$ | $\mathbf{3.97}_{\pm0.09}$ |
| Injection(25%) | $0.80_{\pm0.15}$ | $0.77_{\pm0.12}$ | $\mathbf{3.42}_{\pm0.11}$ |
| Injection(50%) | $0.14_{\pm0.10}$ | $0.04_{\pm0.05}$ | $\mathbf{3.26}_{\pm0.15}$ |
| Injection(100%) | $0.07_{\pm0.08}$ | $0.02_{\pm0.05}$ | $\mathbf{1.68}_{\pm0.18}$ |

## D.4 DIVERSE SUB-OPTIMAL DATASET

Beyond injecting noise into expert demonstrations, we also considered datasets of diverse quality sub-optimal demonstrations. Our decision to focus on noisy expert trajectories stems from the specific challenge we aim to address. Unlike offline-RL, which typically assumes access to reward-labeled trajectories and aims to improve performance from sub-optimal data, our setup considers reward-free offline data where noise degrades originally near-optimal demonstrations. This setting better reflects our goal of studying how exploration can help enhance skill learning when clean supervision is unavailable. To test whether robustness holds across different data qualities, we conducted additional experiments in the Kitchen domain using three dataset types: expert (return $= 4$), medium (return $\approx 2$), and random (collected from a uniformly random policy). As summarized in Table D.3, SISL consistently outperforms other methods across all dataset types, further validating its robustness to data sub-optimality.

Table D.3: Final performance on the Kitchen environment with diverse quality of offline datasets.

| Dataset | SPiRL | SiMPL | SISL |
|---------|-------|-------|------|
| Expert | $3.11_{\pm 0.33}$ | $3.40_{\pm 0.18}$ | $\mathbf{3.97}_{\pm 0.09}$ |
| Medium | $2.62_{\pm 0.27}$ | $3.17_{\pm 0.26}$ | $\mathbf{3.77}_{\pm 0.21}$ |
| Random | $0.07_{\pm 0.08}$ | $0.02_{\pm 0.05}$ | $\mathbf{1.68}_{\pm 0.18}$ |

# E  Computational Complexity

Table E.1 summarizes SISL's computational complexity, showing that the meta-train time increases by about 16% relative to SiMPL, which we view as a modest overhead given the performance gains. For a fair comparison, we keep the amount of training and interaction identical to SiMPL as described in Section 5.1, so the number of samples and policy updates does not exceed that of SiMPL.

Table E.2 summarizes the complexity of SISL ablations. "SISL w/o $P_{\mathcal{B}_{\text{off}}}$" and "SISL w/o $\mathcal{B}_{\text{off}}$" do not perform reward model training and only add skill refinement compared to SiMPL, increasing computation time by about 10%. In contrast, "SISL w/o $\pi_{\text{imp}}$", "SISL w/o $\mathcal{B}_{\text{on}}$", and full SISL additionally perform reward model training, incurring an additional increase of about 3% in computation time. In particular, "SISL w/o $\pi_{\text{imp}}$" differs from full SISL by only 2.3% in training time, indicating that introducing $\pi_{\text{imp}}$ does not lead to a substantial increase in computation cost.

Thus, the 16% overhead mainly arises from skill refinement and reward model training rather than additional policy learning. This overhead is modest in light of the performance gains: as shown in Fig. 6, SiMPL does not improve under high-noise settings even with extended training, whereas SISL continues to improve, justifying this computational overhead. Importantly, considering that the main goal of meta-RL is adaptation to unseen tasks, SISL does not require additional computational cost during the fine-tuning phase and maintains similar computation time as SiMPL in the meta-test phase while still achieving superior performance compared to baseline algorithms.

Table E.1: Comparison of total training time for SiMPL and SISL in Kitchen(Expert).

| Algorithm | Training time (h) | $\Delta$ time vs. SiMPL (%) |
|---|---|---|
| SiMPL | 42.16 | - |
| SISL | 48.96 | +16.1 |

Table E.2: Comparison of total training time for SISL ablation variants in Kitchen(Expert).

| Algorithm | Training time (h) | $\Delta$ time vs. SiMPL (%) |
|---|---|---|
| SISL w/o $\mathcal{B}_{\text{off}}$ | 46.23 | +9.6 |
| SISL w/o $P_{\mathcal{B}_{\text{off}}}$ | 46.71 | +10.7 |
| SISL w/o $\pi_{\text{imp}}$ | 47.98 | +13.8 |
| SISL w/o $\mathcal{B}_{\text{on}}$ | 48.81 | +15.7 |

## F    FURTHER ANALYSIS RESULTS FOR THE SISL FRAMEWORK

In this section, we provide a more detailed analysis and visualization results of the skill refinement process in the SISL. It includes the evolution of the mixing coefficient $\beta$ in Section F.1, visualization of the refined skills and the corresponding skill sequence visualization in Section F.2, task representation in Section F.3, learning stability of the reward model during the meta-train phase in Section F.4, and comparison of skill trajectory with baseline algorithms in Section F.5.

### F.1    EVOLUTION OF THE MIXING COEFFICIENT $\beta$

Fig. F.1 illustrates the evolution of the mixing coefficient $\beta$ during the meta-train phase across all environments and noise levels. Initially, low $\beta$ values reflect reliance on offline datasets for skill learning, particularly in high-return environments like Kitchen and Office, where training starts with $\beta$ values close to zero. This approach prevents performance degradation by avoiding early dependence on lower-quality online samples, while ensuring gradual and stable changes in the skill distribution. As training progresses, the quality of online samples improves, leading to a gradual increase in $\beta$, which leverages online data for skill refinement. For offline datasets with higher noise levels, $\beta$ converges to higher values. Consequently, SISL learns increasingly effective skills as training proceeds, achieving superior performance on unseen tasks, underscoring the importance of SISL's ability to dynamically balance the use of offline and online data based on dataset quality.

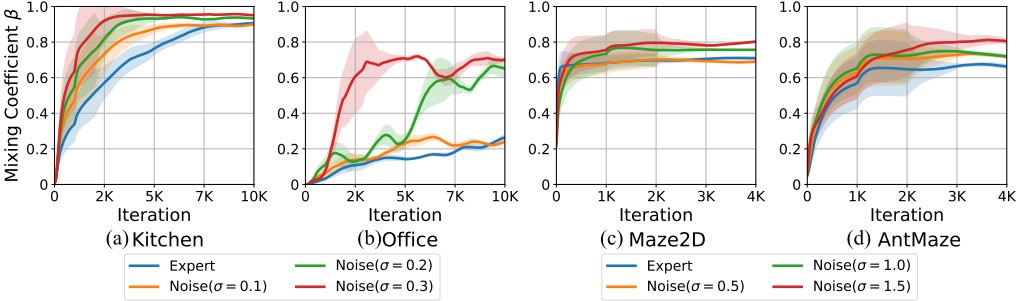

Figure F.1: The evolution of the mixing coefficient $\beta$ during the meta-train phase

### F.2    ADDITIONAL VISUALIZATIONS OF REFINED SKILLS

Here, we present additional visualization results for SISL in the Kitchen and Maze2D environments. Fig. F.2 illustrates the results in the Kitchen environment ($\sigma = 0.3$) after training is completed. On the left, the t-SNE visualization shows the skill representation $z \sim \pi_{h,\theta}$, while the right side highlights the distribution of skills corresponding to each subtask in the t-SNE map and how these skills solve subtasks over time in the Kitchen environment. In the t-SNE map, clusters of markers with the same shape but different colors indicate that identical subtasks share skills across different tasks. From the results, it is evident that the skills learned using the proposed SISL framework are well-structured, with representations accurately divided according to subtasks. This enables the high-level policy to select appropriate skills for each subtask, effectively solving the tasks. Furthermore, while SiMPL trained on noisy data often succeeds in only one or two subtasks, SISL progressively refines skills even in noisy environments, successfully solving most given subtasks.

Fig. F.3 illustrates how the high-level policy utilizes refined skills obtained at different meta-train iterations (0.5K, 2K, and 4K) during the meta-test phase to solve a task in Maze2D ($\sigma = 1.5$). When using skills trained solely on the offline dataset (meta-train iteration 0.5K), the agent failed to perform adequate exploration at meta-test iteration 0K. Even at meta-test iteration 0.5K, the noise within the skills hindered the agent's ability to converge to the target task. In contrast, after refining the skills at meta-train iteration 2K, the agent successfully explored most of the maze during exploration, except for certain tasks in the upper-left region, and achieved all meta-test tasks by iteration 0.5K. Finally, using skills refined at meta-train iteration 4K, the agent not only explored almost the entire maze at meta-test iteration 0K but also completed all meta-test tasks by iteration 0.5K. Additionally, trajectories generated with refined skills showed significantly reduced deviations and shorter paths compared to those using noisy skills. Overall, the results in Fig. F.3 highlight the importance of SISL's skill refinement process in ensuring robust and efficient performance.

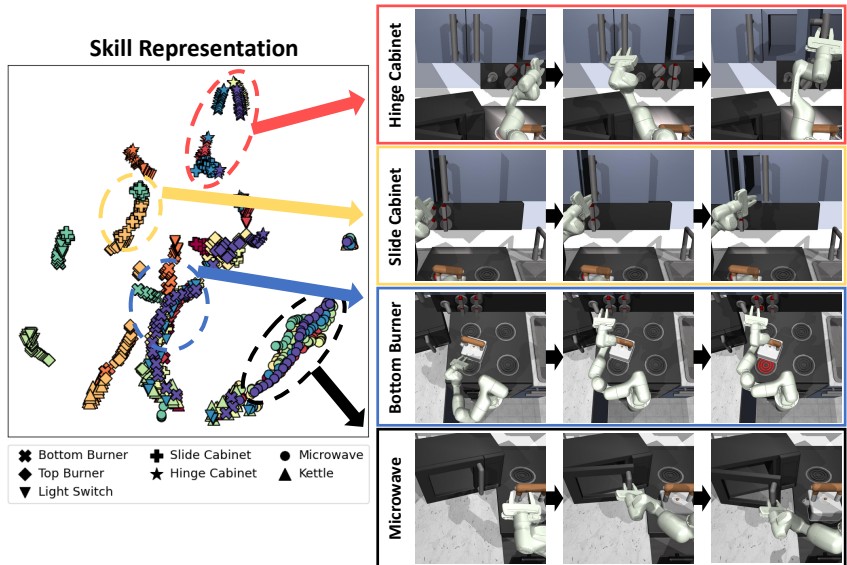

Figure F.2: t-SNE visualization of skill representations (left) and refined skill trajectories for various subtasks (right) in Kitchen ($\sigma = 0.3$). In the skill representation, marker colors denote tasks, while marker shapes indicate subtasks.

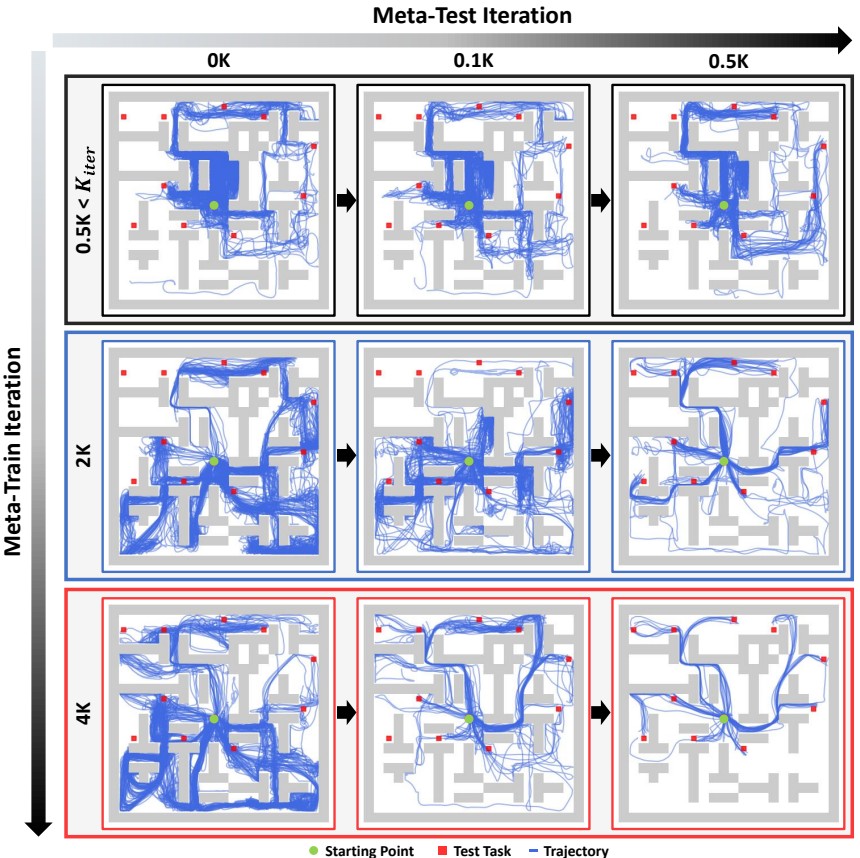

Figure F.3: Illustration of trajectories of refined skills during the meta-test phase in Maze2D ($\sigma = 1.5$), across various training and test iterations.

### F.3 IMPROVEMENT IN TASK REPRESENTATION THROUGH SKILL REFINEMENT

Fig. F.4 illustrates the effect of skill refinement on task representation through t-SNE visualizations of task embeddings $e^{\mathcal{T}} \sim q_e$ in the Kitchen environment ($\sigma = 0.3$), with different tasks represented by distinct colors. In Fig. F.4 (a), the task encoder is trained using fixed skills directly derived from noisy demonstrations. The noisy skills obstruct the encoder's ability to form clear task representations, making task differentiation challenging. This limitation highlights why, in SiMPL, relying on skills learned from noisy datasets can sometimes result in poorer performance compared to SPiRL, which focuses on task-specific skill learning.

Conversely, Fig. F.4 (b) presents the t-SNE visualization when the task encoder is trained during the meta-train phase with refined skills. The improved skills enable the encoder to form more distinct and task-specific representations, facilitating better task discrimination. This improvement allows the high-level policy to differentiate tasks more effectively and select optimal skills for each, thereby enhancing meta-RL performance. These findings demonstrate that the proposed skill refinement not only improves the low-level policy but also significantly enhances the task encoder's ability to represent and distinguish tasks, contributing to overall performance improvements.

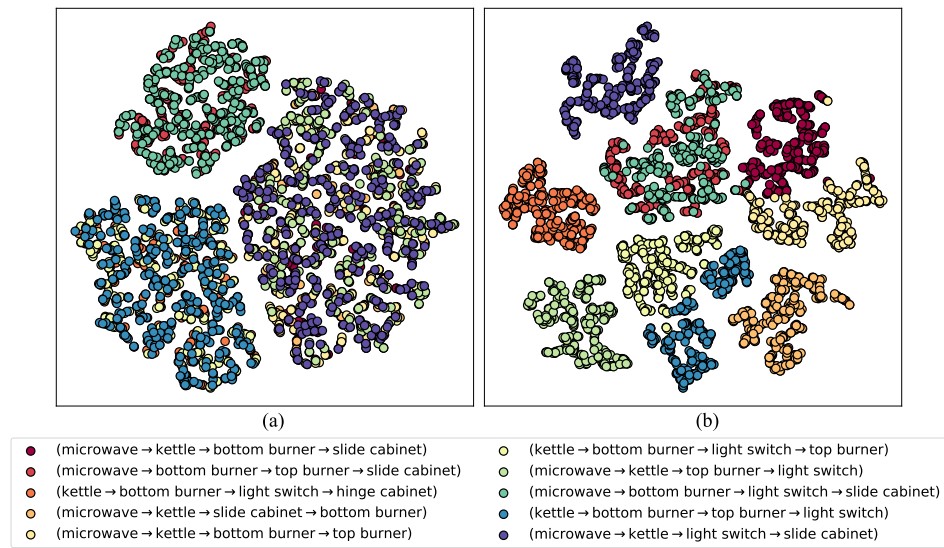

Figure F.4: t-SNE visualization of task representations in the Kitchen environment ($\sigma = 0.3$): (a) Using only the noisy offline dataset, (b) Proposed SISL trained with refined skills

### F.4 STABILITY OF REWARD MODEL LEARNING

Unlike offline-RL, which typically relies on reward-labeled datasets, skill-based RL operates on reward-free offline data. In SISL, only the skills are learned from the noisy offline trajectories, while the reward model and the high-level policy are trained online using clean, noise-free training tasks, as in SiMPL. As shown in Eq. (4), the reward model is trained using transitions sampled from $\mathcal{B}_{\mathrm{imp}}^i$ and $\mathcal{B}_{\mathrm{on}}^i$, which contain only online trajectories collected by interacting with training task $i$. Since these buffers contain no offline data, the reward model is unaffected by offline noise. To further support this point, we present Table F.1, which shows the MSE of the reward model remains consistently low across different offline noise settings in the Kitchen environment. These results indicate that the reward model maintains high accuracy and continues to support effective relabeling even when offline skill pretraining is noisy.

Table F.1: MSE of reward model under different noise levels in the Kitchen environment.

|  | Kitchen(Expert) | Kitchen($\sigma$=0.1) | Kitchen($\sigma$=0.2) | Kitchen($\sigma$=0.3) |
| --- | --- | --- | --- | --- |
| MSE | $0.005_{\pm 0.001}$ | $0.005_{\pm 0.001}$ | $0.004_{\pm 0.001}$ | $0.005_{\pm 0.001}$ |

### F.5 COMPARISON OF SKILL TRAJECTORY VISUALIZATIONS

To visually demonstrate the skill difference between the baseline algorithm and SISL, we have added a visualization comparison of the skill trajectories for SPiRL, SiMPL, and SISL. Specifically, Fig. F.5 and F.6 present the skill trajectory visualizations of the microwave-opening and bottom-burner control subtasks in the Kitchen($\sigma = 0.3$). The SPiRL and SiMPL fail to complete these subtasks due to noise in the learned skills, either do not succeed in opening the microwave door or fail to properly reach the bottom-burner switch. In contrast, SISL successfully grasps and opens the microwave door and correctly manipulates the bottom-burner. These results highlight the substantial impact of noise in offline demonstrations on subtask performance, and demonstrate that SISL progressively refines skills even in noisy environments, successfully solving most given subtasks.

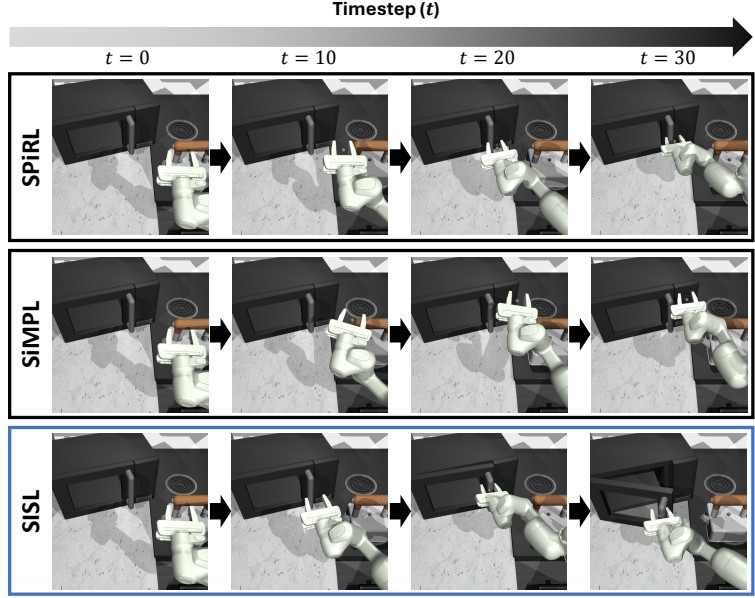

Figure F.5: Comparison of skill trajectory visualizations for the microwave-opening subtask in Kitchen($\sigma = 0.3$) across algorithms.

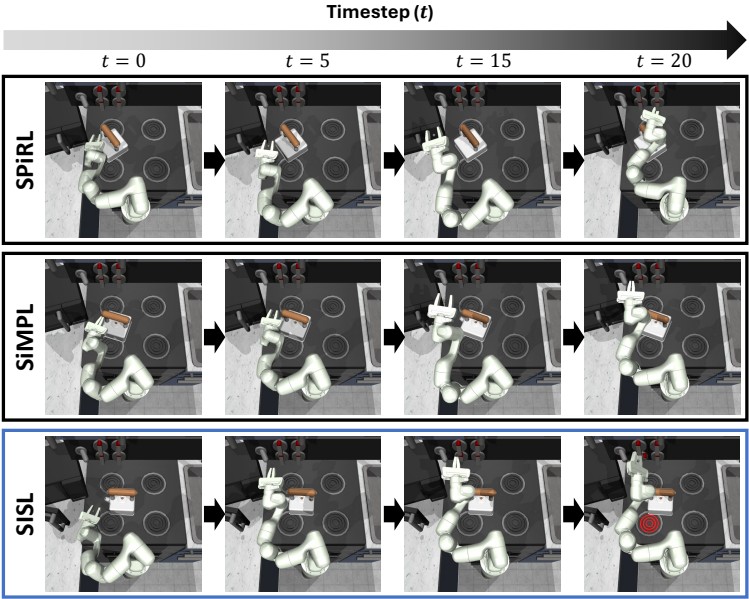

Figure F.6: Comparison of skill trajectory visualizations for the bottom-burner control subtask in Kitchen($\sigma = 0.3$) across algorithms.

## G    ADDITIONAL ABLATION STUDIES

In this section, we conduct additional ablation studies for Kitchen and Maze2D environments across all noise levels. These studies include component evaluation and SISL's skill refinement-related hyperparameters discussed in Section G.1, prioritization temperature $T$ in Section G.2, the KLD coefficient $\lambda_{\text{imp}}^{\text{kld}}$ for the skill-improvement policy in Section G.3, additional component evaluation on RND and re-initialization in Section G.4, skill refinement interval $K_{\text{iter}}$ in Section G.5, and comparison with Goal-Conditioned RL in Section G.6.

### G.1    COMPONENT EVALUATION

Fig. G.1 presents comprehensive results across all noise levels from the component evaluation in Section 5.5. While improvements are modest under conditions with high-quality offline datasets, such as Expert and Noise ($\sigma = 0.1$) for Kitchen, and Expert and Noise ($\sigma = 0.5$) for Maze2D, notable performance gains are still observed. The significant degradation observed in the absence of the skill-improvement policy $\pi_{\text{imp}}$ highlights its crucial role in mitigating minor noise and discovering improved paths. For higher noise levels, such as Noise ($\sigma = 0.2$), Noise ($\sigma = 0.3$) for Kitchen, and Noise ($\sigma = 1.0$), Noise ($\sigma = 1.5$) for Maze2D, excluding the online buffer or skill prioritization via maximum return relabeling ($\mathcal{B}_{\text{on}}$ or $P_{\mathcal{B}_{\text{off}}}$) caused significant performance drops, emphasizing the importance of maximum return relabeling in SISL. Additionally, relying solely on the online buffer without utilizing the offline dataset led to performance deterioration, demonstrating the offline dataset's value in addressing meta-test tasks involving behaviors not available during meta-train. Beyond these specific findings, most trends align with the results discussed in the main text, further validating the effectiveness of SISL's components across different noise levels.

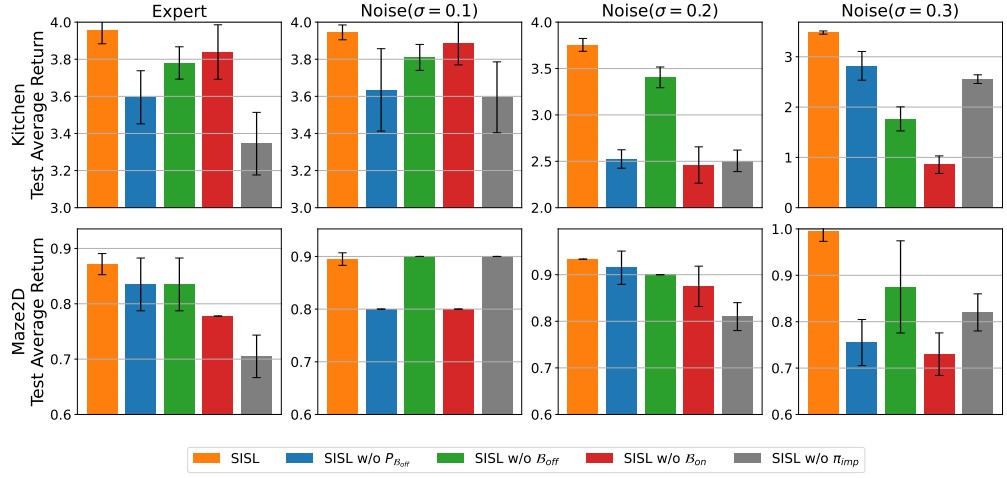

Figure G.1: Component evaluation across all noise levels in Kitchen and Maze2D

### G.2    PRIORITIZATION TEMPERATURE $T$

The prioritization temperature $T$ regulates the balance between sampling from online and offline buffers. Fig. G.2 shows performance across different noise levels in Kitchen and Maze2D environments as $T$ varies. Low $T$ values lead to excessive sampling from high-return buffers, while high $T$ approximates uniform sampling, diminishing the prioritization effect. Both environments experienced degraded performance at $T = 0.1$ and $T = 2.0$, highlighting the importance of proper tuning. In the Kitchen environment (maximum return = 4), $T = 1.0$ achieved the best performance across all noise levels, whereas in the Maze2D environment (maximum return = 1), $T = 0.5$ was optimal. This difference occurs because environments with lower max returns exhibit smaller gaps between low- and high-return buffers, reducing the effect of prioritization. Based on these findings, we suggest a practical guideline: select $T$ roughly in proportion to the environment's maximum achievable return. This approach offers a principled starting point for tuning $T$ without requiring an extensive hyperparameter sweep, thereby improving usability and reproducibility. Accordingly, we set the best-performing hyperparameter values as defaults for each environment.

## G.3 KLD COEFFICIENT $\lambda_{\text{imp}}^{\text{kld}}$

The KLD coefficient $\lambda_{\text{imp}}^{\text{kld}}$ regulates the strength of the KLD term between the skill-improvement policy and the action distribution induced by the prioritized online buffer $\mathcal{B}_{\text{on}}^i$ for each task $\mathcal{T}^i$. Fig. G.3 illustrates performance variations with $\lambda_{\text{imp}}^{\text{kld}} \in [0, 0.001, 0.002, 0.005]$.

In the Kitchen environment, $\lambda_{\text{imp}}^{\text{kld}} = 0.005$ performed best for Expert and Noise ($\sigma = 0.1$), while $\lambda_{\text{imp}}^{\text{kld}} = 0.002$ and $\lambda_{\text{imp}}^{\text{kld}} = 0.001$ were optimal for Noise ($\sigma = 0.2$) and Noise ($\sigma = 0.3$), respectively. At lower noise levels, the high-level policy benefits from quickly following high-return samples, whereas at higher noise levels, focusing on exploration to discover shorter paths becomes more advantageous. For the Maze2D environment, performance was consistent across $\lambda_{\text{imp}}^{\text{kld}} = 0.001, 0.002$, and $0.005$, with only minor variations observed. However, when $\lambda_{\text{imp}}^{\text{kld}} = 0$, removing the KLD term resulted in significant performance degradation across all noise levels in both Kitchen and Maze2D environments. This highlights the necessity of guidance from high-return samples for effectively solving long-horizon tasks. Based on these results, we selected the best-performing hyperparameter values as defaults for each environment.

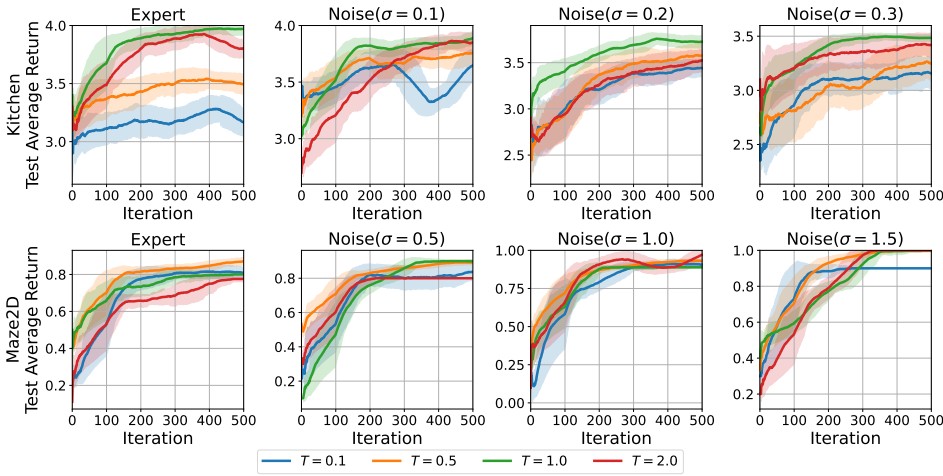

Figure G.2: Impact of the prioritization temperature $T$ across all noise levels in Kitchen and Maze2D

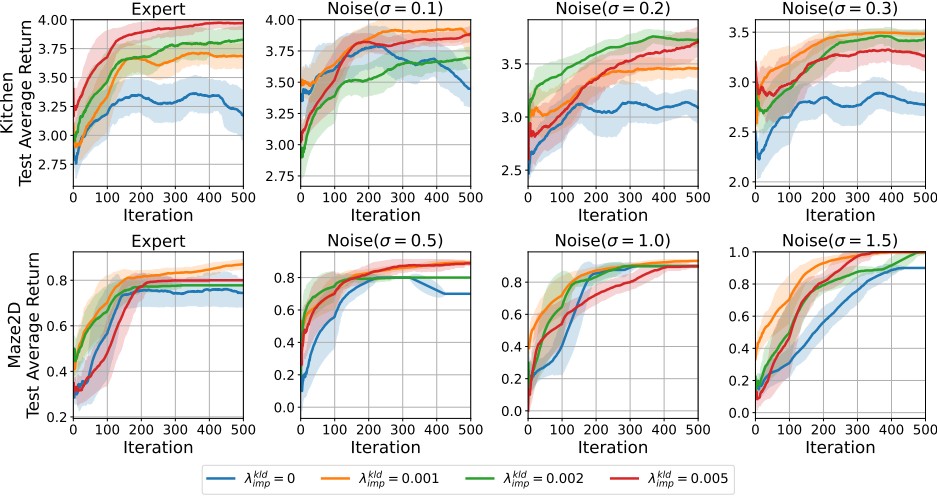

Figure G.3: Impact of the KLD coefficient $\lambda_{\text{imp}}^{\text{kld}}$ across all noise levels in Kitchen and Maze2D

### G.4 ADDITIONAL COMPONENT EVALUATION

Following the ablation of SISL's core components in Section G.1, we performed additional ablations on the remaining components that could affect performance. These evaluations are conducted in the Kitchen environment using both a Expert and a high-noise condition Noise($\sigma = 0.3$), and the results are shown in Table G.1.

The SISL w/o RND is a variant that excludes RND from skill-improvement policy $\pi_{\text{imp}}$ training and relies solely on SAC. The results show performance degradation compared to full SISL, with especially large drops under high-noise conditions where exploration capability is critical. This suggests that RND encourages exploration toward rare or less frequently visited states, enabling the discovery of more diverse and useful trajectories.

The SISL w/o Re-Initialize refers to a variant that continues training the high-level policy $\pi_h$ without re-initializing it at each $K_{\text{iter}}$ during the meta-training process. In SISL, the skill model is periodically updated every $K_{\text{iter}}$ steps, which effectively changes the environment dynamics observed by the high-level policy. Continuing to train the same high-level policy across different skill sets introduces non-stationarity, leading to instability. To address this, we reset both the policy parameters and the buffer at each skill update so that the high-level policy can re-learn from scratch under a new, stable MDP defined by the updated skills. This technique is consistent with practices in continual and safe reinforcement learning, where re-initialization is often used to manage sudden changes in task dynamics (Kim et al., 2023; Kong et al., 2024). To empirically validate this decision, we conducted ablation experiments on high-level policy re-initialization. The results show that removing re-initialization degrades performance, supporting the necessity of this design choice.

Table G.1: Comparison of SISL with/without RND and re-initialization.

| Dataset | SISL | SISL w/o RND | SISL w/o Re-Initialize |
|---|---|---|---|
| Kitchen(Expert) | $3.97_{\pm 0.09}$ | $3.90_{\pm 0.14}$ | $3.11_{\pm 0.22}$ |
| Kitchen($\sigma = 0.3$) | $3.48_{\pm 0.07}$ | $3.14_{\pm 0.10}$ | $0.41_{\pm 0.11}$ |

### G.5 SKILL REFINEMENT INTERVAL $K_{\text{iter}}$

As shown in Fig. 6, the high-level policy $\pi_h$ in the Kitchen environment typically requires at least 1K iterations within each interval to improve and converge, implying that if $K_{\text{iter}}$ is too small, skill refinement is triggered before $\pi_h$ has sufficiently adapted to the current skill library. Based on this observation, we used $K_{\text{iter}} = 2000$ as the default setting in the Kitchen environment. To validate this intuition more rigorously, we conducted hyperparameter search over $K_{\text{iter}} \in \{100, 500, 1000, 2000, 5000\}$ in the Kitchen ($\sigma = 0.3$) setting. The results confirm our hypothesis: with $K_{\text{iter}} = 100$, $\pi_h$ is updated before it can meaningfully exploit the skills, leading to a substantial performance drop. Conversely, when $K_{\text{iter}} = 5000$, $\pi_h$ converges well but skill refinement happens too infrequently, reducing the overall performance gain. These findings support the reasoning behind our chosen interval.

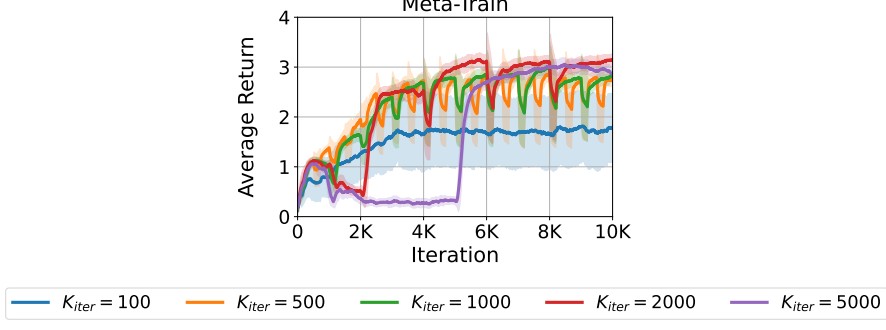

Figure G.4: Impact of the skill refinement interval $K_{\text{iter}}$ in Kitchen($\sigma = 0.3$)

## G.6 COMPARISON WITH GOAL-CONDITIONED RL

We first note that standard goal-conditioned RL (GCRL) assumes explicit, environment-provided goals and single-task learning, whereas in our meta-RL setting no goals are given and the agent must infer task while learning a policy that generalizes across many tasks. Despite these differences, to provide an empirical reference point, we modify our environments to explicitly provide goals and evaluate a representative GCRL method, HER (Andrychowicz et al., 2017) with SAC, trained directly on the test tasks without offline data. As presented in Table G.2, HER achieves notably lower performance than SISL in Maze2D and fails to make progress in AntMaze, where the success rate remains at zero. Even with explicit goals, these sparse-reward long-horizon tasks appear difficult to solve without pretrained skills, which helps contextualize the advantages of our framework.

Table G.2: Final performance of goal-conditioned RL (HER) given a target goal.

| Environment | SAC+HER | SISL |
|---|---|---|
| Maze2D(Expert) | | $0.87_{\pm 0.05}$ |
| Maze2D($\sigma = 0.5$) | | $0.89_{\pm 0.03}$ |
| Maze2D($\sigma = 1.0$) | $0.37_{\pm 0.09}$ | $0.93_{\pm 0.05}$ |
| Maze2D($\sigma = 1.5$) | | $0.99_{\pm 0.02}$ |
| AntMaze(Expert) | | $0.81_{\pm 0.08}$ |
| AntMaze($\sigma = 0.5$) | | $0.82_{\pm 0.05}$ |
| AntMaze($\sigma = 1.0$) | $0.00_{\pm 0.00}$ | $0.60_{\pm 0.02}$ |
| AntMaze($\sigma = 1.5$) | | $0.41_{\pm 0.01}$ |

