# OpenReview forum: "Self-Improving Skill Learning for Robust Skill-based Meta-Reinforcement Learning"
_ICLR.cc/2026/Conference — ICLR 2026 Poster_

### Official Review · Reviewer_LUR5 · 2025-10-18

**Soundness:** 3
**Presentation:** 3
**Contribution:** 2
**Rating:** 6
**Confidence:** 2

**Summary:**

This paper addresses a critical challenge in skill-based meta-RL: performance degradation when learning skills from noisy or suboptimal offline demonstration datasets.

The authors propose a new framework, SISL, designed to be robust to such data imperfections. SISL's novelty lies in two key contributions:
- Decoupled Skill Self-Improvement
- Skill Prioritization via Maximum Return Relabeling

**Strengths:**

- **Significance**: The paper tackles a highly significant and practical problem: the sensitivity of skill-based (meta-)RL algorithms to noisy offline demonstration data. As the field moves toward using large, imperfect, real-world datasets, methods that can "denoise" and "self-improve" upon this data are critical. This work provides a strong solution to this problem.

- **Clarity and Presentation**: The paper is a clear. It is well-written, logically structured, and supported by high-quality figures that provide both conceptual intuition (Fig. 1, 3) and compelling qualitative evidence (Fig. 7, F.2).

- **Empirical Rigor and Performance**: The experimental evaluation is good. The authors demonstrate SISL's superiority across four challenging long-horizon environments against a comprehensive set of relevant baselines.

**Weaknesses:**

- **Complexity and Computational Cost**: The SISL framework introduces several new components that add to the computational load: training the skill-improvement policy, the RND networks, and the reward model, plus periodically re-training the entire skill library.

- **Periodic Re-initialization**: The design choice to re-train the skill library and re-initialize the high-level policy is a key part of the algorithm. The ablation in G.4 shows that not re-initializing is catastrophic, which makes sense due to non-stationarity. However, this periodic "reset" might be disruptive, and the paper does not explore the sensitivity to the frequency itself.

**Questions:**

- **Sensitivity to $K_{iter}$**: The periodic re-training of the skill model and re-initialization of $\pi_h$ every $K_{iter}$ steps is a critical design choice. The ablation in G.4 proves re-initialization is necessary, but how sensitive is the algorithm to the value of $K_{iter}$? What happens if re-training is too frequent (e.g., $K_{iter}=100$) or too infrequent (e.g., $K_{iter}=5000$)?

- **Necessity of Decoupled Policies**: Why is the decoupling of $\pi_h$ and $\pi_{imp}$ strictly necessary? Could a single high-level policy not perform both roles?

---

> ### Author Response · Authors · 2025-11-20
>
> We appreciate the reviewer’s comments, which helped us refine both the presentation and the analysis. Below, we provide detailed responses to each question and clarify the corresponding revisions in the manuscript.
>
> **Weakness 1 (Computational complexity):**
>
> We appreciate the reviewer’s insightful comments. As noted, **Appendix E provides a detailed analysis of SISL’s computational complexity, showing that meta-train time increases by approximately 16\% compared to SiMPL, which we believe is a modest overhead given the resulting performance gains**. Although SISL introduces an additional improvement policy $\pi\_{\text{imp}}$, we ensured a fair comparison by keeping the total amount of training and interaction identical to SiMPL, as described in Section 5.1. In each iteration, half of the tasks use $\pi\_h$ and the other half use $\pi\_{\text{imp}}$, so the total number of samples and policy updates does not increase. The remaining 16\% overhead arises primarily from skill refinement and reward model training rather than additional policy learning.
>
> **Weakness 2/Question 1 (Impact of skill refinement interval $K\_\text{iter}$):**
>
> We appreciate the reviewer’s thoughtful suggestion regarding an ablation study on the skill refinement interval $K\_{\text{iter}}$. As shown in Fig. 6 of the paper, the high-level policy $\pi_h$ exhibits a clear pattern of performance improvement followed by convergence within each interval, and in the Kitchen environment it typically requires at least 1K iterations to reach a stable point. This implies that if $K\_{\text{iter}}$ is set too small, skill refinement would occur before $\pi_h$ has sufficiently adapted to the current skill library, preventing effective learning. Based on this observation, we used $K\_{\text{iter}}=2000$ as the default setting in the Kitchen environment.
>
> To validate this intuition more rigorously, we conducted the reviewer’s suggested hyperparameter search over $K\_{\text{iter}}\in\\{100,500,1000,2000,5000\\}$ in the Kitchen ($\sigma=0.3$) setting. We have included these results in Appendix G.5 and briefly summarized them in Section 5.5. The results confirm our hypothesis: with $K\_{\text{iter}}=100$, $\pi\_h$ is updated before it can meaningfully exploit the skills, leading to a substantial performance drop. Conversely, when $K\_{\text{iter}}=5000$, $\pi\_h$ converges well but skill refinement happens too infrequently, reducing the overall performance gain. These findings support the reasoning behind our chosen interval, and we thank the reviewer for prompting this useful analysis.
>
> **Question 2 (Necessity of decoupled policy):**
>
> We appreciate the reviewer’s question regarding the separation of $\pi\_h$ and $\pi\_{\mathrm{imp}}$. The two policies serve fundamentally different purposes and therefore must be learned independently. The high-level policy $\pi\_h$ selects among already learned skills to maximize return on each task, and because it operates solely within the current skill library, it cannot reliably discover new or improved behaviors. In contrast, the improvement policy $\pi\_{\text{imp}}$ is designed to perturb trajectories near the offline data distribution and actively search for better behaviors that expand and refine the skill library. This functional and structural difference makes a dedicated $\pi\_{\text{imp}}$ essential.
>
> To validate this empirically, we conducted the ablation “SISL w/o $\pi\_{\mathrm{imp}}$” in Section 5.5, where skill refinement is performed using samples collected only by $\pi\_h$. This variant showed substantial performance degradation across all noise levels. These results confirm that **without $\pi\_{\mathrm{imp}}$, the system cannot move beyond the limitations of the initial skill distribution**, making it unable to counteract demonstration noise or discover improved trajectories.
>
>
> Once again, we thank the reviewer for their detailed feedback and insightful suggestions. We hope our clarifications help address the raised concerns and further highlight the contribution of our work.

---

> > ### Comment · Reviewer_LUR5 · 2025-11-25
> >
> > I am satisfied with the rebuttal, which has strengthened the paper. I am leaning towards acceptance; however, given the inherent complexity of the method, I will not be raising my score further. I am maintaining my rating of 6: Marginally above acceptance.

---

> > > ### Author Response · Authors · 2025-11-26
> > >
> > > Dear reviewer LUR5
> > >
> > > We sincerely thank the reviewer for the positive feedback. We fully agree that our method involves additional complexity, and at the same time our ablation studies aim to show that each component is essential for learning skills from noisy datasets. We are very grateful for the time and effort the reviewer has devoted to evaluating our work.

---

### Official Review · Reviewer_SPY1 · 2025-10-23

**Soundness:** 3
**Presentation:** 2
**Contribution:** 3
**Rating:** 4
**Confidence:** 3

**Summary:**

The paper addresses the topic of skill-based meta-learning in reinforcement learning. It points out existing skill learning methods relying on offline demonstrations often suffer from suboptimal data quality. The proposed Self-Improving Skill Learning (SISL) framework shows robustness to noisy demonstrations by introducing a skill improvement policy and performing skill refinement through trajectory prioritization based on returns.

**Strengths:**

The paper adopts perturbations, a representative technique in meta-learning, where target tasks are unseen by the skill-based reinforcement learning agent. This approach constrains the learning process to remain close to the demonstration manifold, thereby facilitating effective skill acquisition.

**Weaknesses:**

The SISL framework appears to be quite complex and computationally expensive. For example, it involves maintaining multiple buffers that serve similar functions. The proposed method also does not seem to be specifically tailored to address the meta-learning problem. Baselines such as SPiRL and SiMPL learn skills solely from offline demonstrations, whereas SISL additionally collects online data, making the comparison unfair.

**Minor**

Appendix B should be moved to the main body of the paper. The figures and their captions are positioned too close to the main text, which reduces readability. Figure captions are brief and could be improved to provide clearer explanations of the content.

**Questions:**

(1) How does the method avoid overfitting to the source tasks when evaluating trajectories based on returns?

(2) Please compare the skills extracted by SISL with those obtained by other baseline methods.

(3) Consider including an additional experiment where Equation (3) is evaluated without the KLD term.

(4) Please report the success rate on the maze environments.

(5) Compare the computational cost across different ablation settings.

(6) Could you clarify why performance increases as $\sigma$ increases in some cases in Table 1?

**Minor**

In Figure 2, which algorithm is being illustrated?

How is the KLD term in Equation (3) computed?

How many offline trajectories generated by the noisy behavior policy are used to solve each task?

---

> ### Author Response · Authors · 2025-11-20
>
> We appreciate the reviewer’s comments, which helped us refine both the presentation and the analysis. To address each question in sufficient detail, we have divided our response into two separate comments and kindly ask for your understanding regarding this format. Below, we provide detailed responses to each question and clarify the corresponding revisions in the manuscript.
>
> **Weakness 1/Question 5 (Computational complexity):**
>
> We appreciate the reviewer’s interest in the computational cost of the proposed components. To address this, we provide a detailed analysis of SISL’s computational complexity in Appendix E. As shown there, **SISL increases computation time by approximately 16\% compared to SiMPL during meta-training, which we believe is a modest overhead given the performance gains.**
>
> Following the reviewer’s suggestion, we further report total training time for the SISL ablations in Appendix E to provide a clearer breakdown of these effects. In Table E.2, “SISL w/o $P\_{\mathcal{B}\_\text{off}}$” and “SISL w/o $\mathcal{B}\_\text{off}$” do not perform reward model training, and instead only add the skill refinement compared to SiMPL. Accordingly, the skill refinement is measured to increase computation time by approximately 10\% relative to SiMPL. In contrast, “SISL w/o $\pi\_\text{imp}$”, “SISL w/o $\mathcal{B}\_\text{on}$”, and full SISL additionally perform reward model training, which together incur an additional increase of about 3\% in computation time. In particular, “SISL w/o $\pi\_\text{imp}$” shows only about a 2.3\% change in training time compared to SISL, indicating that, due to our control environment interaction, introducing $\pi\_\text{imp}$ does not lead to a substantial increase in computation cost. These results show that the additional computational complexity of SISL mainly arises from the skill refinement and reward model training.
>
> **Weakness 2 (Meta-learning formulation):**
>
> We appreciate the reviewer’s concern. Our method is in fact specifically aligned with the objective of meta learning, which aims to train on a set of tasks such that the learned knowledge enables rapid adaptation to unseen tasks. In SISL, skill refinement is performed only on the meta-training tasks so that the resulting skill captures task-general behavior. During meta test, these skills are frozen and only the high-level policy is adapted using a small number of interaction trajectories, allowing fast few-shot adaptation without relearning low-level behaviors. This separation between meta-train skill refinement and meta-test adaptation is precisely what enables SISL to generalize effectively, fully matching the intended purpose of meta learning.
>
> **Weakness 3 (Fair comparison):**
>
> We appreciate the reviewer’s thoughtful comment. To clarify, SISL and SiMPL operate under exactly the same evaluation protocol: both assume access to an offline dataset, both perform online meta-training on the same set of training tasks, and both conduct few-shot adaptation at meta-test. Because **SISL refines skills using only the online samples available within this same meta-training setup, and our refinement procedure does not introduce any additional information beyond what SiMPL already receives**. In this sense, SISL improves learning efficiency within the same problem setting rather than gaining an unfair advantage, similar to how incorporating a learned dynamics model can accelerate RL without compromising fairness as long as sample usage is comparable. Regarding SPiRL, we include it simply as a reference point because SISL adopts the same underlying skill-learning formulation, just as SiMPL does. We hope this clarification resolves the reviewer’s concern about the fairness of the comparison.

---

> ### Author Response · Authors · 2025-11-20
>
> **Question 1 (Overfitting issue):**
>
> We appreciate the reviewer’s thoughtful question regarding potential overfitting of the reward model. In our skill-learning setup, both the offline trajectories and the online training tasks share similar underlying subtasks, such as region-to-region transitions in Ant/Point environments or object-centric subtasks in Kitchen/Office domains. **Because these subtasks define reward semantics consistently across training tasks and offline dataset, so applying the proposed relabeling process does not introduce additional instability from overfitting**. In addition, our softmax-based prioritization forms a distribution rather than relying on a single trajectory, which further mitigates the impact of any minor estimation error. We have clarified this point in Section 5.4. We thank the reviewer again for raising this helpful point.
>
> **Question 2 (Comparison of skill trajectories):**
>
> We appreciate the reviewer's question regarding skill trajectory comparison. To visually demonstrate the skill difference between the baseline algorithm and SISL, we have added a visualization comparison of the skill trajectories for SPiRL, SiMPL, and SISL in Appendix F.5. Specifically, Fig. F.5 and F.6 present the skill trajectory visualizations of the microwave-opening and bottom-burner control subtasks in the Kitchen($\sigma = 0.3$). The SPiRL and SiMPL fail to complete these subtasks due to noise in the learned skills, either do not succeed in opening the microwave door or fail to properly reach the bottom-burner switch. In contrast, SISL successfully grasps and opens the microwave door and correctly manipulates the bottom-burner. These results highlight the substantial impact of noise in offline demonstrations on subtask performance, and demonstrate that SISL progressively refines skills even in noisy environments, successfully solving most given subtasks.
>
> **Question 4 (Success rate of Maze envrionments):**
>
> We appreciate the reviewer’s question. In Maze2D and AntMaze, the environment assigns a return of 1 upon task success. Therefore, the performance values reported in Table 1 directly correspond to success rates. We have clarified this point in Section 5.3 for completeness.
>
> **Question 6 (Performance increases in mild noise):**
>
> We appreciate the reviewer’s observation regarding the performance trends in Table 1, particularly in Maze2D and AntMaze, where performance can improve as the noise level $\sigma$ increases. **As analyzed in Appendix D.1, this behavior arises in mild-noise regimes: increasing noise expands the state coverage of the offline dataset, enabling the model to acquire a more diverse set of skills, which in turn leads to better performance.** Empirically, Maze2D exhibits a 7.3\% increase in state–action coverage at $\sigma=0.5$ compared to the expert dataset, and AntMaze shows a 5.2\% increase under the same noise level. These results support our explanation that mild noise can have a beneficial effect by enriching the data distribution.
>
> **Minor comments:**
>
> **Clarity issues** : We have refined the captions of the figures (Fig. 2, 3, 4) to provide clearer explanations and adjusted the spacing between the figures and the main text to enhance readability. Regarding Appendix B, it offers more detailed implementation information, including network parameters and practical considerations. Due to page limits, the main paper focuses on high-level component descriptions, while the appendix provides the full technical specification.
>
> **Description of Fig. 2** : The prior skill-learning methods referenced in Fig. 2(a) and Fig. 2(b) correspond to the SiMPL algorithm. To avoid ambiguity, we have updated the caption of Fig. 2 to clearly indicate that these are baseline algorithms used for comparison.
>
> **Calculation of KLD in Eq. 3** : The detailed formulation of KLD is provided in Eq. B.3, where the KL regularization term is expressed as
> $\text{KL} = -\sum\_i \mathbb{E}\_{(s\_t, a\_t)\sim \mathcal{B}^i\_\text{on}}[\log \pi\_{\mathrm{imp}}(a\_t \mid s\_t, i)]$.
> In other words, the KL is computed as the average log-likelihood of actions under $\pi_{\mathrm{imp}}$ over samples drawn from $\mathcal{B}^i\_\text{on}$.
>
> **Proportion of offline trajectories** : Since SISL learns skills through a latent representation, it is difficult to explicitly calculate the proportion of offline and online trajectories used during execution. Instead, as shown in Fig. 7 and Appendix F.1, the mixing coefficient $\beta$ provides a clear indication of their relative usage during training process: the model relies more on offline data in the early meta-train phase and gradually shifts toward online samples as their quality improves.
>
>
> Once again, we thank the reviewer for their detailed feedback and insightful suggestions. We hope our clarifications help address the raised concerns and further highlight the contribution of our work.

---

> > ### Comment · Reviewer_SPY1 · 2025-11-27
> >
> > Thank you for providing a clear and detailed rebuttal. Although I still think the method is computationally expensive, it addressed my concerns, and I have accordingly increased my rating to 6.

---

> > > ### Author Response · Authors · 2025-11-27
> > >
> > > Dear reviewer SPY1,
> > >
> > > We sincerely thank you for your positive assessment and for  raising the score. We fully share your concern regarding the complexity, and at the same time we believe that enabling skill learning from noisy data is both important and practically relevant, which motivated us to conduct ablation studies to demonstrate that each component is indeed necessary. We are very grateful for your thoughtful and constructive feedback, which has been highly valuable for improving the paper.

---

### Official Review · Reviewer_jnLn · 2025-11-01

**Soundness:** 2
**Presentation:** 2
**Contribution:** 2
**Rating:** 4
**Confidence:** 3

**Summary:**

This paper targets the sensitivity of skill-based meta-RL methods to noisy offline demonstrations. The proposed method, Self-Improving Skill Learning (SISL), addresses this by dynamically refining the skill library during meta-training. Its core contributions are:

- Decoupled Skill Self-Improvement: A dedicated skill-improvement policy $\pi_{imp}$ is introduced alongside the standard high-level policy $\pi_h$. $\pi_{imp}$ explores near the offline data distribution to find higher-quality trajectories.
- Skill Prioritization via Maximum Return Relabeling: To fuse the noisy $B_{off}$ with the clean $B_{on}$ for skill refinement, SISL trains a reward model using only online data. This model assigns a hypothetical maximum to each offline trajectory. The skill update then samples from $B_{off}$ using a softmax prioritization based on $\hat{G}$, effectively filtering out low-quality data.

Experiments on long-horizon tasks show that SISL substantially outperforms baselines in noisy data regimes.

**Strengths:**

- This work focuses on a key practical problem in meta-RL, where offline demonstrations may be noisy.
- The paper provides strong empirical evidence demonstrating significant performance improvements over all baselines.
- The paper is well-structured. The method is presented logically.

**Weaknesses:**

- The paper claims only "16% more time per iteration", which seems surprisingly low. A SISL iteration appears to involve: (1) rollout with $\pi_h+\pi_l$, (2) rollout with $\pi_{imp}$, (3) $\pi_h$ update, (4) $\pi_{imp}$ update, (5) $\pi_l$ update, and (6) $\hat{R}$ update. In contrast, the baseline presumably only includes steps (1) and (3). It is unclear how the 16% figure was calculated. A more detailed breakdown and a comparison of total training time, not just per-iteration cost, would be more informative.
- The skill refinement interval, $K_{iter}$, is a critical new hyperparameter that balances the stability of the high-level policy $\pi_h$ against the speed of skill refinement for $\pi_l$. However, the paper provides no ablation study for $K_{iter}$, making it difficult to assess the algorithm's sensitivity to this important design choice.
- The "maximum return relabeling" mechanism hinges on a reward model $\hat{R}$ trained on online data. This implies a potential failure mode: in tasks with sparse or complex rewards, training $\hat{R}$ could become unstable, compromising the data prioritization mechanism and overall performance.
- Lack of Clarity in Pseudocode. For example:
    - In Algorithm 1 (Meta-Train), line 14 updates the skill parameters $\phi$ (including $q_{\phi}$). However, the algorithm does not show where the skill encoder $q_{\phi}$ is used for inference on trajectories.
    - In Algorithm 2 (Meta-Test), line 7 updates parameters $\theta$. It is ambiguous whether$\theta$also includes the task encoder $q_{e,\theta}$. If the task encoder is frozen during the meta-test phase (as is typical), it would be clearer to use different notation to distinguish its parameters from the policy/value function parameters being adapted.

**Questions:**

- Figure F.1 shows that the mixing coefficient $\beta$ quickly converges to high values, implying that the online-generated data quality significantly surpasses the offline data. This raises the question: could the offline data $B_{off}$ be discarded altogether? For instance, what if one first trained an expert policy (e.g.,$\pi_{imp}$ trained without the KL-divergence term, acting as a standard RL agent) to collect a new, clean dataset, and then used this dataset for learning $\pi_l$ and $\pi_h$? Would this not be a simpler and potentially lower-cost alternative to the complex relabeling and mixing process?
- Does the framework assume that only optimal trajectories are useful for training the low-level policy $\pi_l$? It is possible that certain suboptimal trajectories, while not useful for the current set of training tasks, might contain skills that are highly beneficial for generalizing to unseen test tasks. The "Maximum Return Relabeling" mechanism prioritizes trajectories based on their estimated maximum return on *training* tasks. Does this design inadvertently suppress these "suboptimal but generalizable" trajectories, thereby weakening the utilization of data that could be crucial for meta-generalization?
- See Weaknesses

---

> ### Author Response · Authors · 2025-11-20
>
> We appreciate the reviewer’s comments, which helped us refine both the presentation and the analysis. To address each question in sufficient detail, we have divided our response into two separate comments and kindly ask for your understanding regarding this format. Below, we provide detailed responses to each question and clarify the corresponding revisions in the manuscript.
>
> **Weakness 1 (Computational complexity):**
>
> We appreciate the reviewer’s insightful comments. As noted, Appendix E provides a detailed analysis of SISL’s computational complexity, showing that meta-train time increases by approximately 16\% compared to SiMPL, which we believe is a modest overhead given the resulting performance gains. Although SISL introduces an additional improvement policy $\pi\_{\text{imp}}$, we ensured a fair comparison by keeping the total amount of training and interaction the same, as described in Section 5.1. **Specifically, in each iteration only half of the tasks use $\pi\_h$ and the other half use $\pi\_{\mathrm{imp}}$, so the total number of samples and policy updates does not exceed that of SiMPL**. The remaining 16\% overhead stems primarily from skill refinement and reward model training rather than from increased policy learning. For completeness, we have also added a more detailed analysis of total training time for SISL in Appendix E, which confirms the same 16\% increase. Since our original per-iteration computation measurements naturally aggregate to the overall training time, the total-time analysis is consistent with and reinforces our initial findings.
>
> **Weakness 2 (Impact of skill refinement interval $K\_\text{iter}$):**
>
> We appreciate the reviewer’s thoughtful suggestion regarding an ablation study on the skill refinement interval $K\_{\text{iter}}$. As shown in Fig. 6 of the paper, the high-level policy $\pi\_h$ exhibits a clear pattern of performance improvement followed by convergence within each interval, and in the Kitchen environment it typically requires at least 1K iterations to reach a stable point. This implies that if $K\_{\text{iter}}$ is set too small, skill refinement would occur before $\pi\_h$ has sufficiently adapted to the current skill library, preventing effective learning. Based on this observation, we used $K\_{\text{iter}}=2000$ as the default setting in the Kitchen environment.
>
> To validate this intuition more rigorously, we conducted the reviewer’s suggested hyperparameter search over $K\_{\text{iter}}\in \\{100,500,1000,2000,5000\\}$ in the Kitchen ($\sigma=0.3$) setting. We have included these results in Appendix G.5 and briefly summarized them in Section 5.5. The results confirm our hypothesis: with $K\_{\text{iter}}=100$, $\pi\_h$ is updated before it can meaningfully exploit the skills, leading to a substantial performance drop. Conversely, when $K\_{\text{iter}}=5000$, $\pi\_h$ converges well but skill refinement happens too infrequently, reducing the overall performance gain. These findings support the reasoning behind our chosen interval, and we thank the reviewer for prompting this useful analysis.
>
> **Weakness 3 (Stability of the reward model):**
>
> We appreciate the reviewer’s insightful question. To address it directly, the reward model used in our maximum return relabeling step is trained exclusively on true online samples, which avoids instability issues. **Moreover, the subtasks that define reward signals are shared across all training tasks and offline dataset used for relabeling, so applying the learned reward model to different tasks does not introduce additional instability.** As reported in Appendix F.4, we evaluated the reward model across all noise levels in the Kitchen environment and observed a consistently low MSE of roughly 0.005. Even under sparse reward conditions, the model reliably identifies when rewards should be positive, confirming its robustness. We thank the reviewer again for raising this point.
>
> **Weakness 4 (Clarification of Pseudocode):**
>
> We appreciate the reviewer’s suggestion to improve the clarity of the pseudocode. In the meta-train phase, the skill encoder $q\_\phi$ corresponds to the encoder used solely for skill learning as defined in Eq. 1 and Eq. B.6, and it is not involved in trajectory rollout or policy optimization, which is why it does not appear explicitly in the pseudocode. We have clarified this distinction in the revised version by explicitly specifying the formulation of the initial skill learning in the pseudocode. Regarding the meta-test phase, as the reviewer correctly pointed out, the task encoder should be fixed to the parameters obtained after training. We have now clarified this by explicitly fixing the parameter $\theta\_\text{final}$ and revising Appendix B.3 and pseudocode accordingly.

---

> ### Author Response · Authors · 2025-11-20
>
> **Question 1 (Necessity of offline dataset):**
>
> We thank the reviewer for raising this point regarding the use of the offline dataset. As suggested, we conducted an experiment in Section 5.5 where the offline dataset $\mathcal{B}\_\text{off}$ is entirely removed and the model is trained solely from online buffer samples (“Ours w/o $\mathcal{B}\_\text{off}$”). As shown in Fig. 8, **this leads to a substantial performance drop even when the offline data are noisy.** Although online data eventually become higher in quality, their improvement relies critically on the KL regularization in Eq. 3. Since the prioritized online buffer $\mathcal{B}\_\text{on}$ is initialized from the offline data, the KL regularization encourages $\pi\_{\text{imp}}$ to explore near the offline data distribution during the early phase of training. Without this guidance, online rollouts fail to make progress on long-horizon tasks, resulting in severe degradation. This demonstrates that, despite noise, leveraging the offline dataset remains essential for effective learning.
>
> **Question 2 (Consideration of suboptimal trajectories):**
>
> We appreciate the reviewer’s thoughtful comment regarding the use of suboptimal trajectories. As the reviewer notes, our method indeed assumes that higher-return trajectories are generally more informative for skill refinement, and we therefore prioritize them during sampling. At the same time, we fully agree that suboptimal trajectories can still contribute valuable structural information. For this reason, our framework does not rely solely on the highest-return trajectories in $\mathcal{B}^i\_\text{on}$; instead, we jointly incorporate the offline dataset $\mathcal{B}\_\text{off}$ through a softmax-based weighting scheme and dynamically mix the two sources using the coefficient $\beta$. This design ensures that suboptimal but relevant trajectories are still included in training, allowing them to influence skill learning while preventing low-quality samples from dominating. We hope this clarifies that our approach is aligned with the reviewer’s intuition.
>
>
>
> Once again, we thank the reviewer for their detailed feedback and insightful suggestions. We hope our clarifications help address the raised concerns and further highlight the contribution of our work.

---

> > ### Comment · Reviewer_jnLn · 2025-11-28
> >
> > Thank you for the response; most of my concerns have been resolved.
> >
> > I retain a question regarding the robustness of the reward model. The model is trained on online data (which mainly contains high-quality trajectories), yet it is used to relabel B_off, which contains static and potentially diverse suboptimal rollouts. There is a risk that this distributional mismatch could introduce bias during the relabeling process.
> >
> > Could the authors comment on whether explicitly incorporating a broader range of non-high-quality trajectories into the reward model training would improve its generalization? This might help bridge the gap between the online and offline distributions and ensure more reliable relabeling of the noisy offline data.

---

> ### Author Response · Authors · 2025-11-28
>
> Dear reviewer jnLn
>
> We sincerely thank the reviewer for raising this important follow-up question. We agree that using a reward model trained on online training tasks to relabel $\mathcal{B}_{\mathrm{off}}$ can introduce a distributional mismatch, even when the online and offline tasks share common subtasks. In our benchmarks (Maze, Kitchen, Office) rewards are provided only at subtask completion states and are zero elsewhere, so the reward depends mainly on whether a particular subgoal is achieved rather than on the rest of the trajectory. In practice, we observed that the learned reward model assigns positive rewards only at subtask completion states and zero otherwise, even on noisy offline rollouts, which suggests that in this subgoal-centric reward setting the model generalizes reasonably well. At the same time we agree that in a dense-reward setup, where rewards depend more sensitively on the entire trajectory, the distributional mismatch highlighted by the reviewer would become more critical and would require a more careful treatment.
>
> Regarding the suggestion to explicitly incorporate a broader range of non-high-quality trajectories into reward-model training, we agree that this could in principle improve generalization between the online and offline distributions. In the skill-based RL setting, however, the offline data used for skill learning consist of state–action sequences without reward labels (as discussed in Section 3), since the skill data are mainly obtained from external sources rather than collected through an online learning process with known rewards. Thus, to address this issue, **one possible extension would be to deliberately generate suboptimal trajectories in the training tasks by injecting noise and then use these samples to train the reward model on a wider range of behaviors. This would, however, require additional sampling and reduce sample efficiency during training, leading to an inherent trade-off between sample efficiency and reward-model generalization.** In our work, the reward model already performed reliably on offline data, so we did not explore this direction further, but we nonetheless view it as a promising avenue for future extensions, especially in dense-reward environments. We hope this clarification addresses the reviewer’s concern, and we are grateful once again for the thoughtful and technically insightful question.

---

### Official Review · Reviewer_brpU · 2025-11-01

**Soundness:** 4
**Presentation:** 4
**Contribution:** 3
**Rating:** 8
**Confidence:** 4

**Summary:**

The authors propose SISL (Self-Improving Skill Learning), a skill-based meta-RL framework designed to handle noisy offline demonstrations in long-horizon tasks. SISL introduces two key mechanisms: (1) decoupled skill self-improvement, where a dedicated improvement policy explores near the offline data distribution to discover higher-quality rollouts that progressively refine the skill library, and (2) skill prioritization via maximum return relabeling, which uses a learned reward model to assign estimated returns to offline trajectories and reweight samples through softmax prioritization.

**Strengths:**

- The paper is well-motivated and very clearly presented, with very useful illustrations.
- Separating exploitation ($\pi_h$) and skill improvement ($\pi_{imp}$) with self-supervised guidance via prioritized buffers appears to be a novel and elegant contribution.
- The evaluation is sound and detailed, with 4 diverse environments with multiple noise levels, thorough baseline comparisons, and extensive ablations.

**Weaknesses:**

I believe the paper would benefit from comparing against a GCRL baseline with Hindsight Experience Replay (HER) or similar relabeling techniques. This would help demonstrate that SISL's approach to leveraging the offline dataset is superior to existing relabeling methods in both sample efficiency and final performance.

Minor typo:
- Line 279: "addtion"

**Questions:**

Shouldn't $\beta$ in Eq. 7 be task-dependent? I guess that sometimes, different tasks may benefit from different balances between $B_{\text{off}}$ and $B_{\text{on}}$. How sensitive is performance to using a global $\beta$ versus task-specific $\beta^i$?

---

> ### Author Response · Authors · 2025-11-20
>
> We appreciate the reviewer’s comments, which helped us refine both the presentation and the analysis. Below, we provide detailed responses to each question and clarify the corresponding revisions in the manuscript.
>
>
> **Weakness 1 (Comparison with Goal-Conditioned RL):**
>
> We sincerely appreciate the reviewer’s suggestion to include comparisons with goal-conditioned RL (GCRL). We would like to clarify, however, that standard GCRL operates under assumptions that differ fundamentally from those in our meta-RL setting. In GCRL, goals are explicitly provided by the environment and learning is performed within a single task, without requiring generalization. In contrast, **our setting does not supply explicit goals; instead, the agent must infer task identity from collected transitions** while simultaneously learning a policy that generalizes across many tasks. Because of these inherent differences, a direct comparison is not straightforward and was not included initially.
>
> That said, we fully understand the reviewer’s interest and conducted additional experiments to bridge this gap as much as possible. By modifying our environments to explicitly provide goals, we evaluated a representative GCRL method, HER with SAC, trained directly on the test tasks without offline data. We have added these results to Appendix G.6, where HER achieves notably lower performance than SISL in Maze2D and fails to make progress in AntMaze, with the success rate remains at zero. Even with explicit goals, these sparse-reward long-horizon tasks appear difficult to solve without pretrained skills, which helps contextualize the advantages of our framework. We hope this clarification and the additional results address the reviewer’s concern.
>
>
> **Question 1 (Task-wise mixing coefficient):**
>
> We thank the reviewer for the insightful comment regarding the use of a task-specific mixing coefficient $\beta^i$. As the reviewer pointed out, when tasks differ significantly in difficulty or exhibit different maximum return scales, adopting a per-task $\beta^i$ could potentially adjust the offline–online data balance more effectively for each task. In our work, however, all training tasks are constructed to share the same reward scale and maximum achievable return. Under this condition, a global mixing coefficient $\beta$ functions reliably without creating imbalance across tasks. Moreover, because SISL maintains separate prioritized online buffers for each task, the use of a shared $\beta$ does not lead to overfitting toward tasks with higher returns. This structure allows us to keep the mechanism simple while preserving robustness. We hope this explanation clarifies our design choice.
>
> **Minor Typo Correction:**
>
> We thank the reviewer for pointing this out. We have fixed the typo in the revised manuscript.
>
> Once again, we thank the reviewer for their detailed feedback and insightful suggestions. We hope our clarifications help address the raised concerns and further highlight the contribution of our work.

---

### Author Response · Authors · 2025-11-20

We sincerely thank all reviewers for their constructive feedback. Following your suggestions, we have substantially strengthened the manuscript with additional experiments and analyses to better demonstrate the computational cost and stability of our framework. A revised version, with all changes highlighted in blue, has been uploaded. The major updates are summarized below.

**(i) Deeper analysis of computational cost (Section 5.1, 5.3, and Appendix E)**

Since SISL introduces several components, we extended Appendix E beyond the original complexity discussion to include total training time comparisons and a more detailed breakdown of computation cost for SISL and its ablation variants. This analysis clarifies where the additional cost arises and shows that it remains moderate relative to the performance gains.

**(ii) Reward model stability and additional visual comparisons (Section 5.4 and Appendix F.5)**

We now provide a clearer explanation of how the reward model used for relabeling is trained stably and avoids overfitting while performing maximum return relabeling. In addition, we include more fine-grained visualizations of skill trajectories for SISL and baseline methods, which highlight their behavioral differences and illustrate why SISL is more effective.

**(iii) Additional ablations and baselines (Section 5.5, Appendix G.5, and G.6)**

To clarify the effect of the refinement schedule, we added a hyperparameter study on the skill refinement interval $K_{\text{iter}}$ and analyzed its impact on performance. We also incorporated an additional comparison with goal-conditioned RL (GCRL) methods, as suggested by the reviewer, to further strengthen the empirical evidence and robustness of our conclusions.

We believe these revisions address the main concerns raised during review and improve the clarity and completeness of the paper. We are grateful for the reviewers’ guidance, which materially enhanced the manuscript.

---

### Author Response · Authors · 2025-12-02

Dear Area Chair,

Thank you very much for your time and effort in handling our submission. In addition to the reviewer-specific and common responses provided in the rebuttal, we would like to briefly summarize the main contribution of our work and how we addressed the reviewers’ concerns, so that the revisions are easier to follow from your perspective.

---

**Main contribution:** SISL addresses the important problem that existing skill-learning methods are heavily reliant on exeprt demonstrations and therefore fail to acquire useful skills when given noisy demonstrations, and is explicitly designed to robustly learn useful skills even from such noisy data. To this end, SISL decomposes the policy into a high-level meta policy that learns skills and **a skill-improvement policy that explores and refines them, initially exploring near the offline data distribution and progressively shifting toward regions induced by its own collected data so that the agent undergoes a self-improvement skill learning process that discovers increasingly better skills over time**. In addition, SISL employs maximum-return relabelling to predict the maximum achievable return for each training task and estimate the usefulness of each skill across tasks, and then performs prioritized skill learning from both online and offline data, which experimentally yields performance close to that obtained with expert demonstrations even under severe noise and significantly outperforms existing skill-learning mechanisms across most benchmarks, substantiating the effectiveness of the proposed framework.

---

**Summary of rebuttal:** Overall, the reviews agree on the necessity of the SISL architecture and its strong empirical performance, and are generally positive about the contribution. The remaining concerns, which were more about the surrounding analysis than the core algorithm itself, mainly focused on computational complexity (reviewers *jnLn*, *SPY1*, *LUR5*), the stability and potential overfitting of the reward model (reviewers *jnLn*, *SPY1*), and the need for additional comparisons and ablations (reviewers *brpU*, *jnLn*, *LUR5*), including analyses of the reinitialization schedule and skill trajectories. In the revision, we provided a more detailed complexity analysis showing that **SISL increases training cost by about 16\%, which we consider moderate relative to the performance gains achieved under noisy demonstrations**. In addition, we clarified **why the reward model remains stable and does not overfit in our setting**, and we substantially expanded our comparisons and ablations against alternative methods such as GCRL. We believe these additions resolve the key concerns raised in the reviews and significantly improve the clarity and completeness of the paper.

---

Due to an issue with the OpenReview system, we unfortunately did not receive all possible follow-up comments. However, reviewers *jnLn* and *SPY1* were the ones who initially raised the most substantial concerns, and both acknowledged that most of their concerns had been resolved. In particular, **reviewer *SPY1* explicitly stated that they were raising their score to 6**, which is reflected in the rebuttal thread. With reviewer *jnLn*, the follow-up discussion focused on the reward model and why overfitting does not occur in our setting, to which we provided a detailed explanation. Since their overall concerns closely mirrored those of *SPY1*, we believe that, had the discussion continued, it would likely have evolved in a similarly positive direction. For the already positive reviewers *LUR5* and *brpU*, reviewer *LUR5* indicated that most of their concerns were resolved and kept their score, and reviewer *brpU* had relatively few concerns, all of which we addressed, so we expect their positive assessment to be maintained.

In summary, SISL makes an original and practically important contribution by tackling robust skill learning from noisy data, and the rebuttal period allowed us to incorporate more detailed analyses of computational cost as well as additional ablations and baseline comparisons requested by the reviewers. We hope that this summary is helpful for your final decision, and we would like to once again sincerely thank you for the careful and thoughtful evaluation of our work.

---

### Meta-Review · Area_Chair_xpam · 2025-12-24

**Summary:**

This paper proposes Self-Improving Skill Learning (SISL), a skill-based meta-RL framework designed to robustly learn skills from noisy or suboptimal offline demonstrations. The reviewers generally agreed that the problem addressed is important and timely, particularly given the increasing reliance on imperfect large-scale datasets in reinforcement learning. The core idea—decoupling skill exploitation from skill improvement and introducing prioritized skill refinement via maximum return relabeling—was viewed as novel and well motivated, and the empirical results consistently demonstrate strong performance gains over existing skill-based meta-RL baselines in long-horizon tasks under noise.

The main concerns raised during the review process focused on (i) computational complexity and implementation overhead, (ii) the stability and potential bias of the learned reward model used for relabeling, (iii) sensitivity to newly introduced hyperparameters such as the skill refinement interval, and (iv) clarity and fairness of comparisons, including additional ablations and baselines. During the rebuttal and revision period, the authors provided additional analyses, experiments, and clarifications that addressed most of these concerns. Overall, the discussion converged toward a more positive assessment, with multiple reviewers explicitly indicating increased confidence in the work after the rebuttal. Based on the technical contribution, empirical strength, and the quality of responses, I am inclined to recommend acceptance.

**Reviewer Concerns:**

Concerns that were addressed by the rebuttal and revision:

•	Computational complexity and overhead:
The authors provided a detailed breakdown of per-component and total training cost in Appendix. Additional ablations further showed that the main overhead comes from skill refinement and reward model training, and that this cost remains moderate relative to the observed performance gains. Reviewers acknowledged that this analysis satisfactorily clarified the concern.

•	Sensitivity to the skill refinement interval:
The revised paper includes a dedicated hyperparameter study analyzing the impact of different refinement intervals. This directly addressed the concern that the interval is a critical design choice without empirical justification.

•	Fairness of comparisons and additional baselines:
The authors added comparisons with goal-conditioned RL  and clarified the evaluation protocol to ensure fairness with SiMPL. These additions strengthened the empirical section and addressed requests for broader context.


•	Clarity of presentation and pseudocode:
The authors clarified the role of the skill encoder, task encoder, and parameter updates in both meta-training and meta-testing, improving the readability and correctness of the algorithm descriptions.

Concerns that are partially addressed or remain as limitations:

•	Reward model generalization and distributional mismatch:
While the authors provided a reasonable explanation—supported by empirical evidence—that the reward model remains stable in their subgoal-centric, sparse-reward benchmarks, the concern about potential bias under more complex or dense-reward settings remains largely conceptual. The authors appropriately acknowledge this as a limitation and outline it as a direction for future work. This does not undermine the validity of the current results but does limit generality.

•	Overall system complexity:
Although well justified and empirically validated, SISL is undeniably complex, with multiple interacting components and buffers. This concern was mitigated but not eliminated; reviewers ultimately viewed the complexity as acceptable given the problem setting and performance gains.

**Reviewer Scores:**

•	Reviewer brpU:
Original score: 8 (Accept).
The reviewer’s concerns (additional GCRL comparison, clarification of mixing coefficient) were fully addressed, and no negative follow-up comments were raised. I believe this reviewer would maintain a score of 8.

•	Reviewer jnLn:
Original score: 4 (Marginally below acceptance).
This reviewer raised the most substantive technical concerns. After the rebuttal, they explicitly stated that most concerns had been resolved, with only a remaining conceptual question about reward-model robustness. Given the added analyses and clarifications, I believe this reviewer would likely raise their score to around 6, reflecting cautious acceptance.


•	Reviewer SPY1:
Original score: 4, explicitly updated to 6 after the rebuttal.
This change was clearly stated by the reviewer during the discussion. The final score would remain at 6.

•	Reviewer LUR5:
Original score: 6 (Marginally above acceptance).
After the rebuttal, the reviewer expressed satisfaction and indicated a leaning toward acceptance while maintaining the same score due to method complexity. I believe this reviewer would maintain a score of 6.

Overall Recommendation: Accept

---

### Decision · Program_Chairs · 2026-01-26

Accept (Poster)